# AGRO: Adversarial Discovery of Error-prone groups for Robust Optimization

**Bhargavi Paranjape**[1][*] **Pradeep Dasigi**[2]  **Vivek Srikumar**[2,3]  **Luke Zettlemoyer**[1,4]
**Hannaneh Hajishirzi**[1,2]
[1]University of Washington,  [2]Allen Institute of Artificial Intelligence,  [3]University of Utah
[4]Meta AI

## Abstract

Models trained via empirical risk minimization (ERM) are known to rely on spurious correlations between labels and task-independent input features, resulting in poor generalization to distributional shifts. Group distributionally robust optimization (G-DRO) can alleviate this problem by minimizing the worst-case loss over a set of pre-defined groups over training data. G-DRO successfully improves performance of the worst-group, where the correlation does not hold. However, G-DRO assumes that the spurious correlations and associated worst groups are known in advance, making it challenging to apply it to new tasks with potentially multiple *unknown* spurious correlations. We propose AGRO—Adversarial Group discovery for Distributionally Robust Optimization—an *end-to-end* approach that jointly identifies error-prone groups and improves accuracy on them. AGRO equips G-DRO with an adversarial slicing model to find a group assignment for training examples which *maximizes* worst-case loss over the discovered groups. On the WILDS benchmark, AGRO results in 8% higher model performance on average on known worst-groups, compared to prior group discovery approaches used with G-DRO. AGRO also improves out-of-distribution performance on SST2, QQP, and MS-COCO—datasets where potential spurious correlations are as yet uncharacterized. Human evaluation of ARGO groups shows that they contain well-defined, yet previously unstudied spurious correlations that lead to model errors.

## 1 Introduction

Neural models trained using the empirical risk minimization principle (ERM) are highly accurate on average; yet they consistently fail on rare or atypical examples that are unlike the training data. Such models may end up relying on spurious correlations (between labels and task-independent features), which may reduce empirical loss on the training data but do not hold outside the training distribution (Koh et al., 2021; Hashimoto et al., 2018). Figure 1 shows examples of such correlations in the MultiNLI and CelebA datasets. Building models that gracefully handle degradation under distributional shifts is important for robust optimization, domain generalization, and fairness (Lahoti et al., 2020; Madry et al., 2017). When the correlations are known and training data can be partitioned into dominant and rare groups, group distributionally robust optimization (G-DRO, Sagawa et al., 2019) can efficiently minimize the worst (highest) expected loss over groups and improve performance on the rare group. A key limitation of G-DRO is the need for a pre-defined partitioning of training data based on a known spurious correlation; but such correlations may be unknown, protected or expensive to obtain. In this paper, we present AGRO—Adversarial Group discovery for Distributional Robust Optimization—an end-to-end unsupervised optimization technique that *jointly* learns to find error-prone training groups and minimize expected loss on them.

Prior work on group discovery limits the space of discoverable groups for tractability. For example, Wu et al. (2022) use prior knowledge about the task to find simple correlations—e.g. presence of negation in the text is correlated with the *contradiction* label (Figure 1). However, such task-specific approaches do not generalize to tasks with different and/or unknown (types of) spurious correlations.

---

[*]Work done during internship at Allen Institute of Artificial Intelligence. Correspondence to: Bhargavi Paranjape <bparan@cs.washington.edu>

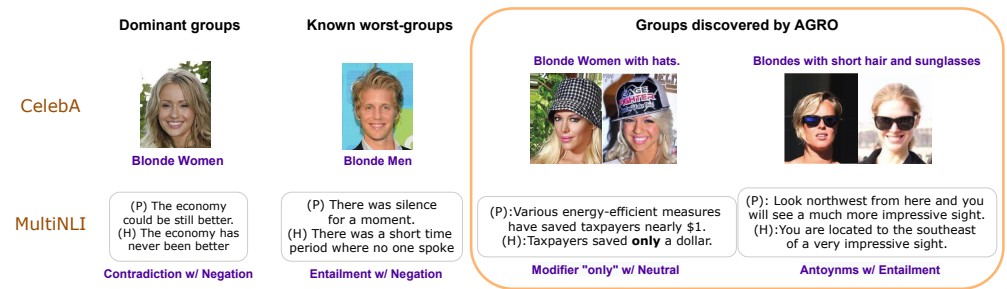

Figure 1: Groups discovered by AGRO on CelebA image classification dataset and MultiNLI sentence pair classification dataset.

Approaches using generalizable features are semi-supervised (Sohoni et al., 2020; Liu et al., 2021) in that they assume access to group information on a held-out dataset. However, obtaining supervision for group assignments is costly and can lead to cascading pipeline errors. In contrast, AGRO is completely unsupervised and end-to-end while making no assumptions about the nature of the task and availability of additional supervision.

To address these challenges in AGRO, we construct a new parameterized *grouper* model that produces a soft distribution over groups for every example in the training data and is jointly trained with the task model. We introduce two key contributions to train this model. Firstly, the grouper model does not make task-specific assumptions about its inputs. Instead, it relies on computationally extracted features from the ERM model including: (a) predictions and mistakes of the ERM model on training and validation instances, and (b) pretrained dataset-agnostic representations and representations fine-tuned on the task data. Secondly, AGRO jointly optimizes the task model and grouper model. We formulate a zero-sum game between the grouper model that assigns instances to groups and the robust model which seeks to minimize the worst expected loss over the set of inferred groups. Specifically, while G-DRO optimizes the robust model to minimize the worst group-loss[1], the grouper model adversarially seeks a probabilistic group assignment such that the worst group-loss is maximized.

On four datasets in the WILDS benchmark (Koh et al., 2021) (MultiNLI, CivilComments, CelebA, and Waterbirds), AGRO simultaneously improves performance on *multiple* worst-groups[2] corresponding to previously characterized spurious correlations, compared to ERM and G-DRO with known group assignment. AGRO also improves worst-group performance over prior approaches that find spurious correlations and groups by $8\%$ on average, establishing a new SOTA for such methods on two of the WILDS datasets. On natural language inference, sentiment analyses, paraphrase detection and common-object classification (COCO), AGRO improves robustness to uncharacterized distributional shifts compared to prior approaches, as demonstrated by gains in out-of-distribution datasets for these tasks. Ablations on different parts of the framework underscore the need for a generalizable feature space and end-to-end optimization. We develop a novel annotation task for humans to analyze the discovered AGRO groups—distinguishing group members from random examples and perturbing them to potentially change model predictions. We find that humans can identify existing and **previously unknown** features in AGRO groups that lead to systematic model errors and are potentially spurious, such as the correlation between antonyms and contradiction in MultiNLI, or the correlation between hats, sunglasses and short hair with non-blondes in CelebA. Our code and models are public[3].

## 2 RELATED WORK

**Group distributionally robust optimization** G-DRO is a popular variant of distributionally robust optimization (DRO) (Ben-Tal et al., 2013; Duchi et al., 2016) that optimizes for robustness over various types of sub-population (group) shifts: label-shifts (Hu et al., 2018), shift in data-sources or domains (Oren et al., 2019), or test-time distributional shifts due to spurious correlations in training

---

[1]Largest group-wise loss in a minibatch

[2]A group of examples where the spurious correlation between feature and label does not hold.

[3]https://github.com/bhargaviparanjape/robust-transformers

data (Sagawa et al., 2019). Santurkar et al. (2020); Koh et al. (2021) introduce a benchmark for robustness to group shifts for measuring progress in this space. There has been significant recent progress on this benchmark using optimization techniques apart from G-DRO — Zhang & Ré (2022); Zhang et al. (2022); Piratla et al. (2021); Kirichenko et al. (2022). However, all these techniques assume that groups or environments are available.

Recent work that focuses on *group discovery* include: (a) GEORGE (Sohoni et al., 2020), which clusters ERM representations to discover groups, (b) JTT (Liu et al., 2021) and LfF (Nam et al., 2020), which use misclassified classes as groups, (c) Nam et al. (2022) use annotations for spurious features on a small subset to build a learned group identifier, and (d) EIIL (Creager et al., 2021) and PGI (Ahmed et al., 2020), which infer groups that maximally violate the invariant risk minimization (IRM) principle (Arjovsky et al., 2019). EIIL is especially relevant to our work: it adversarially maximizes the IRM objective to discover groups. However, the approach is currently limited to two-class problems, and moreover, is challenging to apply for large datasets of the kind we study in this work. Importantly, all these techniques are at least partially supervised, as they assume group memberships are available on a validation set for hyperparameter tuning or model selection. This is a crucial resource, since an ERM-optimized model with access to such a set is also a competitive baseline (Gulrajani & Lopez-Paz, 2020). Another closely related work on distributional robustness for group fairness is ARL (Lahoti et al., 2020), which adversarially trains a network to find instance weights for re-weighting of training examples. One of the core difference between AGRO and ARL is the instance- vs group-based re-weighting. Model errors have been found to be systematic, i.e., occurring in groups (Oakden-Rayner et al., 2020). ARL precludes post-hoc analysis of such patterns due to instance-level modeling.

**Slice Discovery Methods (SDMs)** A related problem to robust optimization is automatic slice discovery (SD)—identifying underperforming data slices for error analysis of deep-learning models. Prior work on SD has used data with tables and metadata (Polyzotis et al., 2019; Sagadeeva & Boehm, 2021). SDMs on unstructured data (Kim et al., 2019; Singla et al., 2021; d'Eon et al., 2022) rely on embedding inputs, dimension reduction, and clustering. Recently, Eyuboglu et al. (2022) propose DOMINO, an error-aware mixture-model based clustering of representations on the evaluation set. While SDMs can help determine patterns in model failures and making them human-comprehensible, they have not been applied for distributional robustness to group (slice) shifts.

## 3 METHOD

### 3.1 PRELIMINARIES

**Problem Setup** We consider the typical image/text classification problem of predicting labels $y \in \mathcal{Y}$ from input $x \in \mathcal{X}$. Training data $\mathbb{D}$ is drawn from the joint distribution $P(\mathcal{X}, \mathcal{Y})$. In state-of-the-art classification models, inputs are typically encoded using multi-layered pretrained transformers (Vaswani et al., 2017; He et al., 2021; Bao et al., 2021) We use $g(x)$ and $h(x)$ to represent input encoding before and after fine-tuning these encoders on $\mathbb{D}$.

**ERM principle** Given a model family $\Theta$ and a loss function $l : \Theta \times \mathcal{X} \times \mathcal{Y} \to \mathbb{R}_+$, empirical risk minimization aims to find a model $\theta \in \Theta$ that minimizes empirical loss over data drawn i.i.d from the empirical distribution $P$. $\hat{\theta}_{ERM} := \operatorname{argmin}_{\theta \in \Theta} \mathbb{E}_{(x,y) \sim P}[l(x, y; \theta)]$. While ERM models achieve high accuracy on i.i.d. evaluation data on average, they often underperform when the test distribution shifts significantly from the training distribution (Madry et al., 2017). ERM also underperforms on a biased sample of the test set where a spurious correlation is absent (Koh et al., 2021).

**G-DRO for Robust optimization** Group-DRO (Sagawa et al., 2019) is a category of distributionally robust optimization (DRO, Duchi et al., 2016), where training distribution $P$ is assumed to be a mixture of $m$ groups, and each training point $(x, y)$ comes from one group $g \in \mathcal{G}$. G-DRO minimizes the empirical worst-group risk $\hat{\mathcal{R}}(\theta)$ i.e. worst (highest) empirical loss over $m$ groups:

$$\hat{\theta}_{DRO} := \arg\min_{\theta \in \Theta} \{ \hat{\mathcal{R}}(\theta) := \max_{g \in \mathcal{G}} \mathbb{E}_{(x,y) \sim p(x,y|g)}[l(x, y; \theta)] \},$$

where $p(x, y|g) \forall g$ is the empirical distribution of training data over groups. With sufficient regularization over $\theta$, G-DRO enables training models that are robust to test-time worst-group distributional shifts. In practice, prior work adopts the efficient and stable online greedy algorithm (Oren et al., 2019)

to update $\theta$. Specifically, worst-group risk $\hat{\mathcal{R}}(\theta) := \max_{q \in \mathcal{Q}} \mathbb{E}_{g \sim q, (x,y) \sim p(x,y|q)}[l(x, y; \theta)]$. The uncertainty set $Q$ are categorical group distributions that are $\alpha$-covered by the empirical distribution over groups $p_{train}$ i.e. $p_{train}(g) := |\mathbb{1}(g_i = g) \forall i \in \mathbb{D}| / |\mathbb{D}|$ and $Q = \{q : q(g) \le p_{train}(g) / \alpha \; \forall g\}$. Effectively, this algorithm up-weights the sample losses by $\frac{1}{\alpha}$ which belong to the $\alpha$-fraction of groups that have the worst (highest) losses, where $\alpha \in (0, 1)$. Details of this algorithm are presented in Appendix A.2. G-DRO is limited to scenarios where both spurious correlations and group memberships are known, which is often not the case.

---

**Algorithm 1:** Online greedy algorithm for AGRO

---

**Data:** $\alpha$; $m$: Number of groups. Minimum $w$ and maximum $W$ weights on group loss; EMA: Expected moving average

**for** $r = 0, 2, ..., R$ **do**

    Initialize historical average group losses $\hat{L}^{(0)}$; historical estimate of group probabilities $\hat{p}^{train(0)}$;

    **for** $t = 1, ..., T_1$ **do**

        $\triangleright$ For group $g \in 1 \dots m$ : $\hat{L}^{(t)}(g) \leftarrow \text{EMA}(\sum_{i=1}^{|B|} P_{i,g}^{(t)} l(x_i, y_i; \theta^{(t-1)}), \hat{L}^{(t-1)}(g))$ ;

        $\hat{p}^{train(t)}(g) \leftarrow \text{EMA}(\sum_{i=1}^{|B|} \frac{P_{i,g}^{(t)}}{|B|}, \hat{p}^{train(t-1)})$;

        $\triangleright$ Sort $\hat{p}^{train(t)}$ in order of **decreasing** $\hat{L}^{(t)}$, top $\alpha$-fraction groups in $\mathcal{A}$ : $q^{(t)}(g) = \frac{1}{\alpha}$ if $g \in \mathcal{A}$ else $w$;

        $\triangleright$ Update model parameters $\theta$ : $\theta^{(t)} = \theta^{(t-1)} - \eta \Delta(\sum_{g=1}^{m} q^{(t)}(g) \sum_{i=1}^{|B|} P_{i,g}^{(t)} l(x_i, y_i, \theta^{(t-1)}))$

    **end**

    **for** $t = 1, ..., T_2$ **do**

        $\triangleright$ For group $g \in 1 \cdot \cdot m$ : $P_{i,g}^{(t-1)} = p(g|f_i; \phi^{(t-1)})$, $L^{(t)}(g) \leftarrow \sum_{i=1}^{|B|} P_{i,g}^{(t-1)} l(x_i, y_i; \theta^{(t)})$,

        $p^{train(t)}(g) \leftarrow \sum_{i=1}^{|B|} \frac{P_{i,g}^{(t-1)}}{|B|}$;

        $\triangleright$ Sort $p^{train(t)}$ in order of **decreasing** $L^{(t)}$, top $\alpha$-fraction groups in $\mathcal{A}$ : $q^{(t)}(g) = \alpha$ if $g \in \mathcal{A}$ else $W$;

        $\triangleright$ Update model parameters $\phi$ : $\phi^{(t)} = \phi^{(t-1)} + \eta \Delta(\sum_{g=1}^{m} q^{(t)}(g) \sum_{i=1}^{|B|} P_{i,g}^{(t-1)} l(x_i, y_i, \theta^{(t)}))$

    **end**

**end**

---

**Error slice discovery** DOMINO (Eyuboglu et al., 2022) shows that systematic mistakes made by models due to reliance on spurious correlations can be exposed via clustering of the model's representations $X$, predictions $\hat{Y}$ and reference labels $Y$. DOMINO learns an error-aware mixture model over $\{X, Y, \hat{Y}\}$ via expectation maximization, finding $m$ error clusters over the evaluation set. Such a clustering can potentially be used for group assignment since the examples in a cluster are coherent (i.e. united by a human-understandable concept and model prediction) and suggestive of a specific feature that the model exploits for its predictions. However, overparameterized neural models often perfectly fit the training data, resulting in zero errors (Zhang et al., 2021), i.e. $Y = \hat{Y}$.

## 3.2 AGRO: ADVERSARIAL DISCOVERY OF GROUP FOR ROBUST OPTIMIZATION

AGRO combines insights from group distributionally robust optimization and error slice discovery, introducing a novel end-to-end framework to accomplish both objectives. We formalize the problem of *group discovery for robustness*, where $g$ is an un-observed latent variable to be inferred during the training process. We replace discrete group memberships $\mathbb{1}(g_i = g); g \in \mathcal{G}$ with a soft-group assignment, i.e. a probability distribution over groups $P_{i,g} := p(g|f_i; \phi); g \in \mathcal{G}$. $P_{i,g}$ refers to the probability that the $i$-th example belongs to the $g$-th group and is realized by a neural network $q$ with parameters $\phi$. This enables modeling co-occurring spurious features and overlapping groups. The input to the *grouper model* $q$ is a high-dimensional feature vector $f_i \in \mathcal{F}$ that can potentially encode the presence of spurious correlations in example $i$ (described in section 3.3).

AGRO jointly trains the parameters of the robust task model, $\theta$, and the grouper model, $\phi$, by setting up a zero-sum game between the robust model and the grouper model. The optimization occurs over $R$ *alternating* game rounds—the **primary round** that updates the task model $\theta$ and the **adversary round** that updates the grouper parameters $\phi$. Parameters that are not optimized in a given round are frozen. In the primary round, the optimizer seeks to minimize the worst-group loss (exactly the G-DRO objective), while in the adversary round, it seeks to find a group assignment that maximizes the worst-group loss. In doing so, the adversary finds a group assignment that is maximally informative for the primary, since it produces a loss landscape over groups that is highly uneven. This forces the primary model, in the next round, to aggressively optimize the worst empirical loss to even out the landscape. With $p_{train}(g) := \sum_{i \in \mathbb{D}} P_{i,g} / |\mathbb{D}|$ and uncertainty set $Q$ is defined as before, the AGRO optimization objective is:

$$\hat{\theta}, \hat{\phi}_{AGRO} := \text{argmin}_{\theta \in \Theta} \{\text{argmax}_{\phi \in \Phi} \{\hat{\mathcal{R}}(\theta) := \max_{q \in \mathcal{Q}(\phi)} \mathbb{E}_{g \sim q, (x,y) \sim p(x,y|q)}[l(x, y; \theta)]\}\}, \quad (1)$$

**Primary Round** In round $r$, the primary classifier finds the best parameters $\theta$ that minimize the worst-group loss based on the current dynamic group assignments provided by $\phi$ in round $r - 1$. Updates to $\theta$ are similar to the online greedy updates used by Oren et al. (2019) i.e. up-weight the loss of $\alpha$ fraction of groups with the highest loss, then minimize this weighted loss.

**Adversary Round** In round $r + 1$, the adversarial grouper updates $\phi$ and learns a soft assignment of groups that *maximizes* the loss of the worst-group (highest loss group). In practice, we adopt the converse of the greedy updates made in the primary round, i.e. down-weight the loss of $\alpha$ fraction of groups with the highest loss, and then maximize this weighted loss.

For stable optimization, we iterate over $T_1$ minibatches of training data to update $\theta$ in the $r^{\text{th}}$ round and iterate over $T_2$ minibatches to update $\phi$ in the $r + 1^{\text{th}}$ round. Algorithm 1 presents the pseudo-code for AGRO. Implementation details, along with hyper-parameter values for $T_1, T_2, \alpha, m$ and $R$, are described in Appendix A.2. In the first primary round, we start with a random group assignment (i.e. random initialization for $\phi$), which amounts to training an ERM model. $\theta$ is initialized with a pretrained transformer-based encoder and MLP classifier. In the first adversary round, we adopt a different initialization for $\phi$, which is explained in Section 4. The grouper model $\phi$ takes as input a feature vector $f_i \in \mathcal{F}$ for each training example $x_i$. Next, we describe our choice of $\mathcal{F}$.

### 3.3 FEATURES FOR GROUP DISCOVERY

Most prior work uses some knowledge about the task (Wu et al., 2022; Gururangan et al., 2018), data collection process (Poliak et al., 2018) or metadata information (Koh et al., 2021) for group discovery. Instead, we develop an unsupervised end-to-end group discovery method that does not rely on any task-specific knowledge. We do so using general purpose features $f_i \in \mathcal{F}$ that can potentially indicate the presence or absence of a spurious correlations.

**ERM features and errors** We use features from weaker ERM models (i.e., under-trained, smaller capacity ones), which have also been used in prior work for automatic group discovery Creager et al. (2021); Sohoni et al. (2020). DOMINO (Eyuboglu et al., 2022) additionally clusters model representations *and* errors to generate coherent error slices on held-out data. However, overparameterized models often perfectly fit training data. To estimate model errors on training data, we apply K-fold cross-validation to get model predictions on held-out training instances in every fold. Specifically, for a training example $x_i$ assigned to the $k$-th fold's held-out set, we compute the following features: the model's fine-tuned representations $h(x_i)$, the reference label $y_i$, and the vector of prediction probabilities over labels $\{p(\hat{y}_i|x_i; \theta_k) \forall \hat{y}_i \in \mathcal{Y}\}$, where $\theta_k$ is the fold-specific ERM classifier.

**Pretrained features** Recent work has shown that large transformer-based models trained on web-scale data learn general features for images which improve on out-of-distribution data (Wortsman et al., 2022a;b). Pretrained models are less likely to encode dataset-specific features in their representations. Following this observation, we also include pretrained representations $g(x_i)$ of the input $x_i$. In sum, the group discovery features $f_i$ for a training example $x_i$ are: the representations $g(x_i)$ and $h(x_i)$, the label $y_i$ and the probabilities $\{p(\hat{y}_i|\theta_k) \forall \hat{y}_i \in \mathcal{Y}\}$.

## 4 EXPERIMENTAL SETUP

**Datasets with known spurious correlations** We evaluate performance on four datasets in language and vision from the popular **WILDS benchmark** (Koh et al., 2021) to study in-the-wild distribution shifts. We select tasks that contain more than one known worst-group and report on the top three known worst-groups (referred to as KWGs) from these datasets, chosen based on the rate of correlation between feature and label. Table 1 briefly describes these known worst-groups by dataset. More details about tasks descriptions, statistics, and pre-processing can be found in Appendix A.3.

**Datasets without known spurious correlations** Our main contribution is a fully unsupervised end-to-end technique for group discovery and robustness, which can be applied off-the-shelf on a new dataset with no additional annotation. To evaluate this, we train AGRO for robustness to spurious correlations in datasets with possibly unknown biases and evaluate its generalization on OOD datasets, where such correlations may not hold. We train models on two datasets from the GLUE benchmark (Wang et al., 2018)—Stanford Sentiment analysis (SST2) and Quora question paraphrase detection (QQP), and a modified version of image classification on the MS-COCO dataset (Lin et al., 2014),

| Dataset | Inputs | Labels | Worst Groups |
|---|---|---|---|
| MultiNLI | Sentences | Entail, Contradict, Neutral | KWG1: Negation-Contradict, KWG2: Lexical overlap - Entail |
| Waterbirds | Image | Waterbird, Landbird | KWG1: Waterbird w/ land background , KWG2: Landbird w/ water backgrounds |
| CelebA | Image | Blonde, Non-Blonde | KWG1: Male Blondes, KWG2: Blondes wearing eyeglasses, KWG3: Blondes w/ big noses |
| Civil Comments | Comment | Toxic, Non-toxic | KWG1: Toxic comment mentioning black, KWG2: LGBTQ and KWG3: other religions |

Table 1: Inputs, labels and known worst-groups in WILDS datasets.

COCO-MOD. We evaluate on out-of-distribution (OOD) datasets for these tasks, like IMDb (Maas et al., 2011) for SST2, PAWS (Zhang et al., 2019) for QQP, and SpatialSense (Yang et al., 2019) for MS-COCO.

**Hyperparameters** For $\theta$, we use transformer-based (Vaswani et al., 2017) pretrained models that have been shown to be robust to rare correlations (Tu et al., 2020), including DeBERTa-Base (He et al., 2020) for text classification and BEIT (Bao et al., 2021) for image classification. $\phi$ is an 2-layer MLP with a softmax activation layer of size $m$ (the number of groups). Appendix A.4 and A.6.3 further detail hyperparameter tuning, particularly for AGRO-specific hyperparameters $\alpha$ and $m$.

**Initializing** $\phi$ A random initialization to grouper $\phi$ can result in a degenerate group assignment, where all examples are assigned the same group. Such an assignment still results in a small proportion of groups (*one* group) having the maximum loss. This reduces to the ERM solution. To mitigate this behavior, we initialize the grouper model's parameters $\phi$ in the first adversary round to predict the same group memberships as the clusters from DOMINO (Eyuboglu et al., 2022), which was described in Section 3.1. Our ablation results in Section 6 show that AGRO improves over directly using DOMINO slices as groups for G-DRO. Details about the pretraining of $\phi$ are in Appendix A.2.

**Model Selection** Unlike prior work on automatic group discovery (e.g, Liu et al., 2021; Creager et al., 2021), we do not assume access to group annotations on the validation set based on a known spurious correlation to tune hyperparameters or to choose the best model checkpoints. We develop a new strategy for model selection based on the learned grouper model $\phi$. We select the best model based on the combined performance of $\alpha$-fraction of the worst-performing groups discovered by AGRO. For a fair comparison with the baselines described below, we similarly use the worst-group available to each baseline. In Section 6, we relax this assumption for all methods (AGRO and baselines) and use KWG1 of each dataset for model selection.

**Baselines** We compare against previous automatic group discovery approaches used with G-DRO:

JTT: Liu et al. (2021) retrain on training data errors made by a weaker ERM model. Model selection is based on errors made on the validation set.

GEORGE: Sohoni et al. (2020) cluster the training and validation sets based on ERM features and loss components. The worst-group on the validation set is used for model selection.

EIIL: Creager et al. (2021) propose to train a group discovery model to adversarially maximize the invariant risk minimization objective. Upon examination of their experimental details, we find that for all large datasets (that we report on), they use labels as groups, since EIIL is hard to scale. We follow the same procedure to compare EIIL with AGRO. For small datasets Waterbirds and SST2, we use EIIL as is. The class with the worst validation performance is used for model selection.

G-DRO: Following (Sagawa et al., 2019), we optimize G-DRO based on a group assignment for a known spurious correlation, and use the same assignment on the validation set for model selection. G-DRO serves as an upper bound on the worst-group performance for this spurious correlation. Specifically, we use KWG1 to run G-DRO on all datasets.

ERM: ERM models trained for longer and with larger batch size already improve worst group performance (see Appendix A.6), making for a strong baseline.

## 5 RESULTS

### 5.1 AGRO IMPROVES ROBUSTNESS TO MULTIPLE SPURIOUS CORRELATIONS

AGRO learns a soft group assignment over training data, with the goal of improving worst-group accuracy over multiple co-occurring correlations. We evaluate this hypothesis on the selected datasets

| Group Accuracy | ERM | G-DRO | JTT | GEORGE | EIIL | **AGRO** |
|---|---|---|---|---|---|---|
| MultiNLI | | | | | | |
| Full evaluation set | 87.0(.04) | 84.8(.05) | 79.6(1.2) | 58.4(.99) | 86.4(.7) | 85.31(.5) |
| KWG1:Negation-Contr. | 75.8(.09) | 84.2(.9) | 71.9(2.0) | 54.3(0.7) | 76.1(1.) | **82.29(.23)** |
| KWG2:Lexical Overlap-Ent. | 45.5(.05) | 43.1(1.0) | 48.2(1.4) | 21.0(1.3) | 47.0(.55) | **51.8(.95)** |
| Waterbirds | | | | | | |
| Full evaluation set | 97.75(1.) | 96.3(.9) | 96.4(1.3) | 97.8(.8) | 96.5(1.3) | 97.7(1.1) |
| KWG1:Waterbirds-Land | 90.2(.7) | 97.7(.8) | 83.5(2.0) | **96.2(1.0)** | 78.8(.8) | 96.2(1.2) |
| KWG2:Landbirds-Water | 97.4(.8) | 93.4(1.2) | 96.1(1.5) | 95.9(1.4) | 84.6(.5) | **96.1(1.0)** |
| CelebA | | | | | | |
| Full evaluation set | 95.5(.4) | 94.0(1.3) | 92.9(1.0) | 94.7(1.5) | 94.5(.91) | 95.59(1.) |
| KWG1:Males-Blonde | 39.6(.23) | 86.8(1.9) | **75.8(.8)** | 57.7(3.0) | 55.5(.5) | 69.2(1.2) |
| KWG2:Eyeglasses-Blonde | 53.8(.88) | 82.7(.8) | 82.7(1.6) | 61.5(2.5) | 78.8(.7 ) | **82.5(.99)** |
| KWG3: Big Nose-Blonde | 69.4(.78) | 92.5(.7) | **88.9(.67)** | 77.8(2.1) | 82.1(.81) | 87.10(1.6) |
| Civilcomments | | | | | | |
| Full evaluation set | 92.9(.4) | 82.0(1.6) | 73.9(1.0) | 90.1(.08) | 88.0(.01) | 91.2(1.9) |
| KWG1:Black-Toxic | 58.3(.55) | 71.7(1.7) | 40.7(2.3) | 52.8(.09) | **88.7(.88)** | 72.0(2.) |
| KWG2:LGBTQ-Toxic | 53.1(.31) | 83.8(.81) | 43.6(3.1) | 64.2(1.) | **78.4(.92)** | 66.2(1.5) |
| KWG3: Other Religions-Toxic | 53.1(.67) | 82.1(.91) | 34.0(1.8) | 60.4(.41) | **84.4(.6)** | 66.7(.8) |

Table 2: Average and WG performance on WILDS datasets. AGRO is better at multiple worse groups and also consistently improves over ERM, unlike prior work. Best KWG performance bolded and second-best underlined. (*) indicates variance across 3 random seed runs.

| MultiNLI | | | | | SST2 | | | | |
|---|---|---|---|---|---|---|---|---|---|
| Dataset | ERM | GEORG. | EIIL | AGRO | Dataset | ERM | GEORG. | EIIL | AGRO |
| PI | 80.27 | 78.27 | 81.53 | 81.52 | SST2 | 92.90 | 87.10 | 66.39 | 92.01 |
| LI | 88.45 | 81.84 | 87.88 | 89.24 | Senti140 | 88.58 | 83.67 | 53.99 | 85.16 |
| ST | 76.75 | 71.21 | 75.93 | 76.29 | SemEval | 83.90 | 89.1 | 72.14 | 88.58 |
| HANS | 74.58 | 66.83 | 67.06 | 73.12 | Yelp | 91.60 | 91.9 | 84.05 | 92.01 |
| WaNLI | 60.82 | 55.94 | 59.86 | 63.12 | ImDB | 85.97 | 85.74 | 64.50 | 86.78 |
| SNLI | 83.21 | 80.75 | 83.00 | 84.78 | Contrast | 85.25 | 83.60 | 56.76 | 86.68 |
| ANLI(R3) | 30.17 | 31.92 | 29.00 | 34.92 | CAD | 90.78 | 89.95 | 58.20 | 90.37 |
| Avg% $\Delta$ | | -4.68 | -2.27 | 3.04 | | | -1.20 | -26.30 | 0.50 |
| QQP | | | | | COCO-mod | | | | |
| Dataset | ERM | GEORG. | EIIL | AGRO | Dataset | ERM | GEORG. | EIIL | AGRO |
| QQP | 91.17 | 88.24 | 82.73 | 91.58 | COCO-mod | 89.14 | 87.52 | 79.28 | 89.26 |
| PAWS | 38.08 | 30.87 | 43.72 | 41.8 | Spatial | 89.89 | 87.25 | 64.4 | 91.33 |

Table 3: Accuracy on out-of-distribution datasets (columns) for 4 tasks with unknown spurious correlations. AGRO improves over ERM by .5-10%, while baselines underperform.

from WILDS (Table 1). Table 2 reports accuracy on the full evaluation set and on multiple worst-groups. Additional results using weaker pretrained models are in the Appendix A.6.

Across datasets, AGRO improves worst-group accuracy by 25.5% on average over the ERM baseline without hurting the overall task accuracy. In particular, it improves across *multiple* known worst-groups for each dataset. On all datasets except CelebA, for KWG1, AGRO gets within 2% of the accuracy obtained by G-DRO with *ground truth* groups. This shows that ARGO's unsupervised group discovery and robust optimization performs nearly as well as using the best manually designed groups. No other baseline consistently improves worst-group accuracy over all tasks. Approaches like GEORGE, JTT and EIIL have mixed results across datasets. Their performance depends on specific properties of the test datasets, such as high spurious correlation ratio (in CelebA) or severe label imbalance (in Civilcomments). These results indicate that AGRO is most suitable for new datasets whose key properties are unknown. Section 5.2 explores this further.

On **MultiNLI**, DeBERTa-based models achieve the best reported worst-group performance by both the ERM model and the G-DRO model (trained using KWG1 assignments). The relative gap between

worst and average accuracy persists with a weaker model like RoBERTa (see Appendix A.6). AGRO has an 8% improvement over the next highest baseline on this task (EIIL). GEORGE, which uses loss components for clustering groups and for model selection, underperforms even on in-domain data, potentially from over-fitting to groups with label noise. The worst group performance of AGRO establishes a new state of the art for group robust optimization approaches that also find groups. On **Waterbirds**, the ERM model finetuned using BEIT has competitive worst-group accuracy, surpassing the performance gap of a weaker model like ViT(see Appendix A.6). We hypothesize that a stronger pretrained model is generally less susceptible to spurious correlations (Wortsman et al., 2022a). On both worst groups, AGRO outperforms ERM and has comparable performance with the next best baseline GEORGE, once again achieving SOTA.

**CelebA** and **Civilcomments** have overlapping groups i.e. images with multiple facial attributes and comments with mentions of multiple demography labels, respectively. Consequently, the performance of G-DRO with ground-truth groups for KWG1 also improves performance on other worst groups. On the Male-Blonde group in **CelebA**, AGRO outperforms the ERM model by 75%. It also outperforms GEORGE and EIIL on this group. Owing to a high rate of spurious correlation in the dataset (male:non-male blondes ratio is 0.02), re-weighting misclassified points (JTT) proves to be an effective grouping mechanism. **Civilcomments** has a significant label imbalance (a ratio of 0.13 for toxic to non-toxic, the lowest for all datasets considered in this work). Consequently, using the label-as-group heuristic (EIIL) performs the best among other baselines, followed by AGRO.

## 5.2 AGRO IMPROVES ROBUSTNESS TO DISTRIBUTION SHIFTS ON NEW TASKS

Since AGRO is fully unsupervised, it can be used off the shelf to improve distributional robustness on a new task. To evaluate this, we apply AGRO to SST2, QQP, and COCO-MOD, datasets where spurious correlations and worst-groups are unknown. Models trained on these datasets are tested on out-of-distribution (OOD) datasets, where spurious correlations learned in-distribution may not hold. In Table 3, we report average and out-of-distribution accuracies achieved by AGRO and two best performing baselines, GEORGE and EIIL. We also report OOD performance for MultiNLI.

A striking observation across tasks and datasets is that baselines generally underperform ERM on general out-of-distribution robustness. We hypothesize that this may be because of over-fitting to noisy groups with high losses. G-DRO can be effective at improving OOD robustness if distributional shifts in the groups correspond to distributional shifts at test time. We hypothesize that the baselines overfit to groups in the in-domain data in unexpected ways due to their simplified grouping heuristics, resulting in poor OOD generalization. On MultiNLI, AGRO outperforms ERM by 3.0% on average across OOD datasets. On COCO-MOD, AGRO outperforms ERM by 2% on the OOD dataset. On SST2, AGRO improves performance over ERM on most datasets by 0.5-5.5% except on Senti140 (Twitter sentiment classification dataset). On QQP, AGRO outperforms other baselines, including ERM. The JTT baseline on QQP gives us the best results on out-of-distribution data, but at the cost of in-domain performance. We hypothesize that AGRO's OOD gains stem from reduced reliance on dataset-specific correlations.

## 6 ANALYSIS

We present detailed analysis to determine which aspects of AGRO contribute the most to its strong unsupervised performance and do a qualitative study of AGRO groups.

**Feature Ablation** Table 4 (right) reports the impact of adding different features used in AGRO on the MultiNLI dataset. We find that using error-aware clustering of features (i.e. DOMINO in Row 3) as group assignments improves worst-group performance compared to clustering based on features alone (i.e. only using $h(x_i)$). Using adversarial group discovery along with these features especially boosts performance on the worst group, underscoring the importance of our end-to-end framework. Finally, adding pretrained features to AGRO gives us the best performance on MultiNLI. Ablation results for Waterbirds are in Appendix A.6.4.

**Model Selection** Table 4 (left) reports the results on using knowledge of worst group during evaluation for model selection. We report worst-group performance(of KWG1) of AGRO and all baselines for MultiNLI by selecting model checkpoints based on that worst group. The worst-group performance of all baselines increases. For AGRO, our selection strategy from Section 4 is as effective as using the known worst group for selection (the results remain unchanged). AGRO and the baselines EIIL and

| Algorithm | AvgAcc | KnownWG |
|---|---|---|
| ERM | 86.63 | 77.96 (+2.9%) |
| JTT | 81.87 | 74.27 (+3.3%) |
| GEORGE | 85.44 | 76.03 (+40.0%) |
| EIIL | 86.38 | 77.45 (1.8%) |
| AGRO | 85.31 | 82.29 (Same) |
| G-DRO | 84.81 | 84.20 (Same) |

| Features | AvgAcc | KnownWG |
|---|---|---|
| ERM | 86.28 | 74.01 |
| Feature clustering | 84.15 | 74.14 |
| DOMINO clustering | 85.65 | 76.82 |
| AGRO | 85.39 | 79.24 |
| AGRO + pretrained features | 85.97 | 80.64 |

Table 4: Improvement in accuracy on WG1 of MNLI when it is used for model selection (left). Ablation of features in $\mathcal{F}$ and ablation for adversarial learning over clustering (right).

| WG1: Qualifiers like "all, only, solely" with Neutral class. |
|---|
| P: Chapter 5 focuses on synergistic combinations of control retrofits on a single unit. |
| H:Chapter 5 focuses ~~solely~~ on synergy (Neutral → Entail) |
| P:but that that's my that's probably my main hobby that sewing and reading books that's about it |
| H:Sewing and reading books are my ~~only~~ hobbies. (N → E) |
| P:Nuns are in their habits and Israeli women in military uniforms. |
| H:~~All~~ Israeli women are in the military, which is why they wear military uniforms. (N → E) |

| WG2: Antonyms in P and H with Contradiction class. |
|---|
| P: Stand behind it. |
| H: Stand so that it is ~~in front~~ **behind** of you. (Contradict → Entail) |
| P: Look for the Cecil Hotel on the western end of the square. |
| H: You can see the Cecil Hotel from the ~~centre~~ **west** of the square. (C → E) |
| P: Get there early in the morning before the crowds arrive |
| H: You should get there shortly ~~after~~ **before** they open to beat the crowds. (C → E) |

Table 5: Perturbation patterns on worst groups discovered in AGRO. Class flips in ()

GEORGE have competitive worst-group accuracies. These observations hold on the Waterbirds data as well (A.6). Our selection strategy can potentially also be used with other group robust optimization techniques, which we will explore in future work.

**Small-scale human study** Appendix A.8 has examples of slices discovered by AGRO on several datasets. The core goal of AGRO is to discover error-prone groups without prior knowledge of spurious correlations. Human expertise is needed to ascertain whether a model exploits any spurious correlations in the discovered slices (Eisenstein, 2022). Soliciting human feedback is challenging since models may exploit spurious correlations that do not correspond to human-interpretable or identifiable features. We design two annotation tasks to obtain feedback indirectly. The first annotation task evaluates **coherency**. Annotators are shown four instances from a group identified by AGRO, along with two other test instances, only one of which is from the same group. The annotators are asked to identify the test instance that belongs to the group. The second annotation task follows the first, where the annotators are asked to perturb the instance chosen to be part of the AGRO group in such a way that the task model's prediction would change. This exercise can implicitly surface the common *feature* humans think exists in the slice. We conduct the human study on 5 slices each from MNLI, SST2 and CelebA. More details of the human annotation setup can be found in Appendix A.7. For Task 1, we find that inter-annotator agreement has a score of 0.7823 Fleiss Kappa. Humans were able to identify the correct group member 70.1% of the time. Some examples of the flips provided by human annotators in Task 2 are presented in Table 5.

## 7 CONCLUSION

We studied the challenge of applying G-DRO to new tasks with *multiple, unknown* spurious correlations and groups and present AGRO—Adversarial Group discovery for Distributionally Robust Optimization— an end-to-end approach that *jointly* identifies error-prone groups and improves model accuracy on these groups. AGRO's adversarially trained grouper model finds a soft group assignment that is maximally informative for G-DRO. Compared to prior work, AGRO is a more effective off the shelf optimization technique for fully-unsupervised group discovery and robustness on new tasks.

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

# A APPENDIX

## A.1 ADDITIONAL RELATED WORK

**Other Robust Optimization techniques** Other approaches for robustness to group distributional shifts include data augmentation and ensemble learning. (Liu et al., 2022; Wu et al., 2022) (for MultiNLI) and (Plumb et al., 2021) (for COCO) use data augmentation. However, they make task-specific assumptions about the types of features that can be spurious to generate additional data. Product-of-experts(PoE)-based (Hinton, 2002)approaches like (Sanh et al., 2020; Mahabadi et al., 2019; He et al., 2019) use a biased model to construct a robust ensemble. The biased model is typically constructed using task-specific information. Since Since AGRO finds error-prone groups that potentially encode model biases, these groups can be used for data-augmentation or PoE approaches in future work.

## A.2 ALGORITHMS AND IMPLEMENTATION DETAILS

**G-DRO** Algorithm 2 presents the online greedy algorithm used in G-DRO. Note that group assignment over training data is based on a predefined spurious correlation. EMA refers to exponential weighted moving average. i.e. $EMA(v1, v2) = \gamma v_1 + (1 - \gamma)v_2$, where $\gamma \in (0, 1)$ and $\alpha \in (0, 1)$. $\mathcal{A}$ is the $\alpha$ fraction of groups with the highest loss. Specifically, based on the group proportions $\hat{p}_{train}^{(t)}$, $\mathcal{A}$ consists of those groups with high loss values that make up $\alpha$-fraction of the dataset. The greedy up-weighting scheme is based on the CVaR-style implementation of G-DRO from Zhou et al. (2021) and Sagawa et al. (2019). They seek to optimize:

$$\hat{\theta}_{G-DRO} := \arg\min_{\theta \in \Theta}\{\hat{\mathcal{R}}(\theta) := \max_{q \in \mathcal{Q}} \mathbb{E}_{g \sim q, (x,y) \sim p(x,y|q)}[l(x, y; \theta)]\}, \tag{2}$$

The uncertainty set $Q$ are group distributions that are $\alpha$-covered by $p_{train}$, which is the empirical distribution of groups in training data. Specifically, $p_{train}(g)$ is the proportion of training examples that belong to group $g$.

$$p_{train}(g) := |\mathbb{1}(g_i = g)\forall i \in \mathbb{D}|/|\mathbb{D}|$$

, and

$$Q = \{q : q(g) \leq p_{train}(g)/\alpha \; \forall g\}$$

In their code, Zhou et al. (2021) recognize that $p_{train}(g)$ can't be exactly computed since optimization occurs in mini-batches. $|\mathbb{1}(g_i = g)\forall i \in \mathbb{D}|$ is therefor estimated based on the number of examples of group $i$ in minibatch (of size B). In order to have an unbiased estimate for $p_{train}(g)$ they maintain an expected moving average $\hat{p}_{train}(g)$. Thus,

$$\hat{p}_{train}(g) \leftarrow \text{EMA}(|\mathbb{1}(g_i = g)\forall i \in \mathbb{B}|/|\mathbb{B}|, \hat{p}_{train}(g))$$

In implementation, the optimization up-weights the sample losses by $\frac{1}{\alpha}$ which belong to the $\alpha$-fraction of groups that have the worst (highest) losses, where $\alpha \in (0, 1)$. This algorithm is presented in Appendix A.2. All the other groups are down-weighted by a minimum group weight, $w$. We set $w = 0.1$ following prior work. We tune $\alpha$ based on known worst-group accuracy on each dataset.

**DOMINO** (Eyuboglu et al., 2022) propose a mixture model that jointly models input embeddings, class labels, and model predictions. This encourages s slices that are homogeneous with respect to error type (e.g. all false positives), while also being coherent with respect to human-interpretable features. The model assumes that data is generated according to the following generative process: each example is randomly assigned membership to a *single* slice among $k$ possible slices according to a categorical distribution $\mathbf{S} \sim \text{Cat}(\mathbf{p}_S)$ with parameters $\mathbf{p}_S \in \{\mathbf{p} \in \mathbb{R}_+^k : \sum_{i=1}^k p_i = 1\}$. Given membership in slice $j$, the embeddings are distributed normally as $Z|S^{(j)} = 1 \sim \mathcal{N}(\mu^{(j)}, \Sigma^{(j)})$ with parameter mean $\mu^{(j)} \in \mathbb{R}^d$ and covariance $\Sigma^{(j)} \in \mathbb{S}_{++}^d$ (the set of symmetric positive definite $d \times d$ matrices), the labels vary as a categorical $Y|S^{(j)} = 1 \sim \text{Cat}(\mathbf{p_{(j)}})$ with parameter $\mathbf{p(j)} \in \{\mathbf{p} \in \mathbb{R}_+^c : \sum_{i=1}^c p_i = 1\}$, and the model predictions also vary as a categorical $Y|S^{(j)} = 1 \sim \text{Cat}(\hat{\mathbf{p}}_{(\mathbf{j})})$ with parameter $\hat{\mathbf{p}}(\mathbf{j}) \in \{\hat{\mathbf{p}} \in \mathbb{R}_+^c : \sum_{i=1}^c \hat{p}_i = 1\}$. The embedding, label, and prediction are all independent conditioned on the slice.

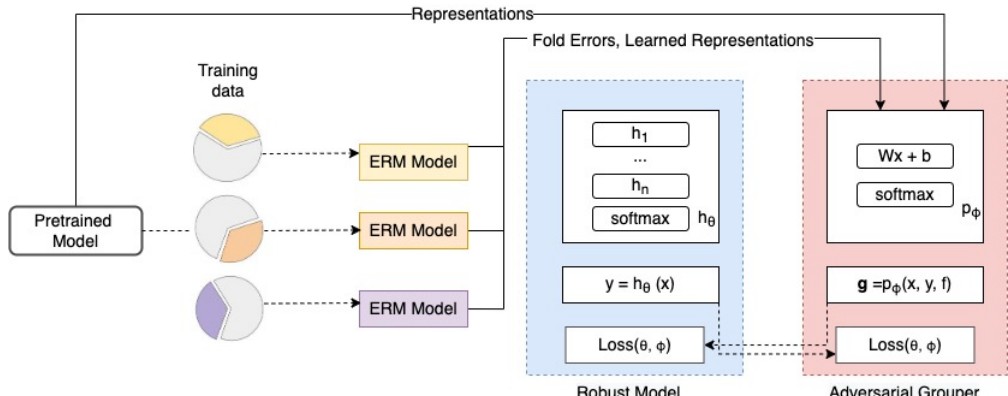

Figure 2: Adversarial Group discovery for Distributionally Robust Optimization (AGRO) framework

The mixture model is parameterized by $\phi = [\mathbf{p}_S, \{\mu^{(j)}, \Sigma^{(j)}, \mathbf{p_{(j)}}, \hat{\mathbf{p}}_{(j)}\}_{j=1}^k]$. The log-likelihood over the validation dataset $D_v$ is given as follows and maximized using expectation-maximization:

$$l(\phi) = \sum_{i=1}^n \log \sum_{j=1}^k P(S^{(j)} = 1) P(Z = z_i | S^{(j)} = 1) P(Y = y_i | S^{(j)} = 1)^\gamma P(\hat{Y} = h_\theta(x_i) | S^{(j)} = 1)^\gamma$$

, where $\gamma$ is a hyperparameter that balances the trade-off between coherence and under-performance and $h_\theta(x_i)$ are the class logits predicted by the ERM model on example $x_i$. $\gamma$ is set to 1 for all the datasets considered in this work.

**AGRO**    The AGRO optimization objective is as follows:

$$\hat{\theta}, \hat{\phi}_{AGRO} := \arg\max_{\phi \in \Phi} \{\arg\min_{\theta \in \Theta} \{\hat{\mathcal{R}}(\theta) := \max_{q \in \mathcal{Q}(\phi)} \mathbb{E}_{g \sim q, (x,y) \sim p(x,y|q)}[l(x, y; \theta)]\}\}, \qquad (3)$$

The uncertainty set $Q$ are group distributions that are $\alpha$-covered by $p_{train}$, similar to G-DRO. However, $p_{train}(g)$ is the proportion of training examples that belong to group $g$.

$$Q = \{q : q(g) \le p_{train}(g)/\alpha \; \forall g\}$$

. In the case of AGRO, $p_{train}(g)$ is computed based on a group assignments probabilities output by the adversary model $\phi$ trained in the previous round. Specifically, for minibatch $\mathbb{B}$

$$\hat{p}_{train}(g) \leftarrow \text{EMA}(\sum_{j=1}^{|\mathbb{B}|} \frac{p(g|f_j, \phi^{(t-1)})}{|B|}, p_{train}(g))$$

, where $p(g|f_j, \phi^{(t-1)})$ is the probability of example $j$ in the minibatch belonging to group g, which is estimated by the adversary model $\phi$.

**Primary Round**    To optimize $\theta$, we up-weights the mini-batch sample losses by $\frac{1}{\alpha}$ which belong to the $\alpha$-fraction of groups that have the worst (highest) losses, where $\alpha \in (0, 1)$. All the other groups are down-weighted by a minimum group weight, $w$. We set $w = 0.1$ following prior work. We tune $\alpha$ based on known worst-group accuracy on each dataset.

**Secondary Round**    We compute probability distribution over $m$ groups for instance $i$, $P_i^{(t)}$, using the grouper model of the previous iteration, $\phi^{(t-1)}$. This soft group assignment is then used to compute weighted loss and soft group count for the minibatch in iteration $t$. As described in 3.2, in order to train the grouper model in the adversary round, we **reverse** the greedy weighting procedure adopted in G-DRO. Therefore, if $\mathcal{A}$ is the $\alpha$ fraction of groups with the **highest** loss, the $\alpha$-fraction of groups in $\mathcal{A}$ are down-weighted by $\alpha$, while all the other groups are up-weighted by a maximum group weight, $w$. We set $W = 1.0$, which we find works for all the datasets considered in this work.

**Iterative optimization** We set the number of rounds to $R = 1$, since we found no further gains with more rounds of AGRO and high computational cost. The computational cost for further rounds is considerable since the feature space $\mathcal{F}$ needs to be re-computed for the new robust model $\theta$. The optimization process is further simplified by setting $T1$ and $T2$ to a certain number of epochs. Since we first begin with training the primary model in round 0, we start with a random group assignment, which basically amounts to training a weaker ERM model for $T1$ epochs. After this we train the adversary for $T2$ epochs. After one round of training the primary and adversary model, we again train the primary model for the same number of epochs that we use to train a strong ERM model baseline that we compare against. Hyperparameters $m$ and $\alpha$ are tuned for each dataset separately, which we explain in greater detail in Appendix A.4.

**Initializing $\phi$** In the adversary round, the grouper model learns a soft assignment of groups that maximizes the loss of the worst-group (highest loss group). In practice, this means down-weighting the loss of $\alpha$ fraction of worst-loss groups and then maximizing this weighted loss. If $\phi$ is initialized randomly, the maximization results in a degenerate solution — all examples are forced into one group. This solution trivially maximizes the loss over a small $\alpha$ fraction of groups, since all examples are forced into those $\alpha$ groups. In order to prevent this degeneracy, we provide an initialization to $\phi$ that already creates an uneven loss distribution over groups. DOMINO clusters examples based on model errors, labels and representations. Consequently, they possess two properties that can be a good initializing for $\phi$ — (a) Coherence, slices that have similar features are grouper together, and (b) Homogeneous errors, slices with the same predicted and reference label are generally grouper together. Thus, we train the grouper MLP parameters $\phi$ to predict cluster memberships provided by DOMINO on training data (which we obtain using K-fold training). Specifically, we minimize the KL-divergence between the probability distribution over groups generated by the grouper model and the probability distribution over groups predicted by the generative mixture model trained by DOMINO. We pre-train the grouper model for 10 epochs, with a batch size of 256.

**Constructing $\mathcal{F}$** Figure 2 is a visualization of the feature generation pipeline used to create features for the grouper model $f_i \in \mathcal{F} \forall x_i \in \mathbb{D}$.

---

**Algorithm 2:** Algorithm 1: Online greedy algorithm for G-DRO from Oren et al. (2019).

---

**Data:** $\alpha$; $m$: No. of groups
Initialize historical average group losses $\hat{L}^{(0)}$; historical estimate of group probabilities $\hat{p}^{train(0)}$;
**for** $t = 1, ..., T$ **do**
    $\triangleright$ For group $g \in 1, , m$;
    $\hat{L}^{(t)}(g) \leftarrow \text{EMA}(l(x_i, y_i; \theta^{(t-1)}) : \mathbf{g}_i = g, \hat{L}^{(t-1)}(g))$;
    $\hat{p}^{train(t)} \leftarrow \text{EMA}(\text{No. samples of each group in B}, \hat{p}^{train(t-1)})$;
    $\triangleright$ Sort $\hat{p}^{train(t)}$ in order of **decreasing** $\hat{L}^{(t)}$, sorted group indices in $\pi$;
    $\mathcal{A}$ is the top $\alpha$-fraction groups with highest loss;
    $q^{(t)}(g_{\pi_i}) = \frac{1}{\alpha}$ if $g \in \mathcal{A}$ else $w$;
    $\triangleright$ Update model parameters $\theta$;
    $\theta^{(t)} = \theta^{(t-1)} - \frac{\eta}{|B|} \sum_{i=1}^{|B|} q^{(t)}(g_{\pi_i}) \Delta l(x_i, y_i, \theta^{(t-1)})$
**end**

---

## A.3 TRAINING AND EVALUATION DATASETS

### A.3.1 WILDS BENCHMARK

We experiment with 4 datasets from the WILDS benchmarkKoh et al. (2021) with multiple potential spurious correlations. Each is described in greater detail here. For more detail on each task, we refer the reader to original paper that used them for G-DRO — Sagawa et al. (2019).

**MultiNLI** Williams et al. (2017) task is to classify a pair of sentences as one of entailment, neutral, contradiction. We follow the train/dev/test split in (Sagawa et al., 2019), which results in 206,175 training examples. (Gururangan et al., 2018) identified that negation words in the second sentence are spuriously correlated with the contradiction labels. Accordingly, examples are grouped based on if the second sentence contains any negation word in the list (nobody, nothing, no, never). Test/validation data follows same group-class distribution as train. In addition to the negation spurious correlation that is tested in the standard benchmark, we also test for the correlation between lexical overlap and the entailment label. To construct groups for this correlation, we consider token-level overlap (number

of common token-types between premise and hypothesis normalized by hypothesis length) $> 0.9$ to contain the lexical overlap heuristic. In order to measure robustness to general distributional shifts on MNLI, we also evaluate on the following out-of-distribution datasets following (Wu et al., 2022). The NLI adversarial test benchmark Liu et al. (2020b) that tests for the following heuristics/short cuts that models may exploit: (a) Partial Inputs (PI) Heuristics Gururangan et al. (2018); Liu et al. (2020a); (b) Inter-sentences (IS) Heuristics, specifically the syntactic diagnostic dataset HANS McCoy et al. (2019); (c) Logical Inference Ability (LI) Glockner et al. (2018); Minervini & Riedel (2018); and (d) Stress-test (ST) Naik et al. (2018). We also evaluate on other crowdsourced NLI datasets like SNLIBowman et al. (2015), WaNLILiu et al. (2022) and Adversarial NLI Nie et al. (2019).

**WaterBirds**    WaterBirds task is to predict if an image contains water-bird or a land-bird and contains two groups: majority, minority in ratio $62 : 1$ in training data. The minority group has reverse background-foreground coupling compared to the majority. Test/validation data has equal proportion of each group.

**CelebA**    CelebA Liu et al. (2018) task is to classify a portrait image of a celebrity as blonde/non-blonde, examples are grouped based on the gender of the portrait's subject. The training data has only $1,387$ male blonde examples compared to $200K$ total examples. Test/validation data follows same group-class distribution as train.

**CivilComments**    CivilComments-WILDS Borkan et al. (2019) task is to classify comments as toxic/non-toxic. Examples come with eight demographic annotations: (white, black, LGBTQ, muslim, christian, other religions, male and female) based on if the comment mentioned terms that are related to the demographic. This leads to a significant group overlap, where the same comment will have several demography words at once. Label distribution varies across demographics. In the benchmark train examples are grouped on black demographic, and testing is done on the worst group accuracy among all 16 combinations of the binary class label and eight demographics. In particular, we report on the three worst groups i.e. groups with rarest co-occurrence rate between demography label and class label — LGBTQ-toxic, other religions-toxic and black-toxic.

### A.3.2    OTHER DATASETS

To demonstrate the effectiveness of AGRO on new datasets where spurious correlations may be uncharacterized as of yet, we experiment with two additional text classification tasks and 1 image classification task. Since the worst-group based on a known spurious correlations is not known, we instead measure performance on out-of-distribution data which can contain unknown group-shifts. While AGRO does not target those specific group shifts and is only guaranteed to provide robustness over group distributions that it finds in-domain, we believe that generally reducing reliance on spurious correlations may lead to improvements on other group shifts as well.

**SST2**    SST2 task Socher et al. (2013) from the GLUE NLU benchmark Wang et al. (2018) is to classify 1-sentence movie reviews as positive or negative. SST2 has an equal proportion of labels in training and evaluation sets. For robustness evaluation, we use the same out-of-distribution datasets as those used by (Wu et al., 2021) — twitter sentiment analysis tasks (Go et al.; Asghar, 2016), ImDB sentiment analysis Maas et al. (2011), and counterfactual datasets constructed over ImDB **?** (Contrast) and **?** (CAD).

**QQP**    Quora question paraphrase detection task (QQP) from GLUE is to classify a pair of question extracted from the Quora platform as paraphrase or not-paraphrase. The task consists of 400,000 label-balanced training instances. For robust evaluation, we use (Zhang et al., 2019), a dataset consisting of a group distributional shift that tests for the spurious correlation of lexical overlap and paraphrase label. It consists of questions with the same words but a different word order.

**COCO-MOD**    We modify the COCO (Common Objects in Context)(Lin et al., 2014) image classification dataset into a 7-class problem instead of the more complex 80 object classification problem. We select the following 7 classes: [Tie, Toothbrush, Bird, Frisbee, Tennis racket, Dog, Fork]. These classes are specifically chosen based on the findings of Plumb et al. (2021) (see Table 2). They find that these 7 object classes are often associated with other spurious objects. For example,

| Hyperparameter | Notation | Range of values |
|---|---|---|
| Fraction of worse groups | $\alpha$ | $\{0.1, 0.2, 0.3, 0.4, 0.5\}$ |
| Number of groups | $m$ | $4, 6, 9, 12, 24, 50^*$ |
| Iterations of adversary round | $T2$ | 1,3,5 |
| Learning rate | $\eta$ | $1e-6, 1e-5, 2e-5$ |
| Weight decay | $\lambda$ | $0.0, 1.0, 2.0$ |
| Hidden size of $\phi$ | $\phi_h$ | $64, 128$ |

Table 6: Hyperparameter Sweep values for AGRO. * For a particular dataset, we only sweep over values of $m$ such that there are $> 5\%$ examples on average in a group.

| Dataset | $\alpha$ | $m$ | $T_2$ | $\eta$ | $\lambda$ | $\phi_h$ | $\theta$ | N | B |
|---|---|---|---|---|---|---|---|---|---|
| MultiNLI | 0.2 | 9 | 5 | 2e-5 | 0.0 | DeBERTa-Base | 64 | 15 | 144 |
| CivilComments | 0.2 | 12 | 2 | 2e-5 | 0.0 | DeBERTa-Base | 64 | 15 | 192 |
| CelebA | 0.4 | 12 | 1 | 2e-5 | 0.0 | beit-large-patch16-384 | 64 | 15 | 48 |
| Waterbirds | 0.2 | 6 | 1 | 2e-5 | 0.0 | beit-large-patch16-384 | 64 | 20 | 48 |
| SST2 | 0.4 | 6 | 1 | 1e-6 | 2.0 | RoBERTa-base | 64 | 20 | 128 |
| QQP | 0.4 | 6 | 2 | 2e-5 | 0.0 | RoBERTa-base | 64 | 20 | 128 |
| COCO | 0.4 | 24 | 4 | 2e-5 | 0.0 | vit-base-patch16-224-in21k | 64 | 10 | 128 |

Table 7: Final hyperparameter settings

ties with cats and toothbrushes with people. We apply AGRO on the classification task for these 7 objects to potentially rediscover these correlations.

## A.4 HYPERPARAMETER SEARCH

Hyperparameter search for G-DRO are other baselines are discussed individually in A.5. Here, we focus on hyperparamter search for AGRO. The list of hyperparameters and the range of values used for search can be found in Table 6. Since the hyperparamter space to explore is quite large, we use the following values for other hyperparameters that we keep fixed while doing a linear sweep over a particular hyperparameter. The order of hyperparamter search is in the same order as the rows in Table 6. In Table 7, we report the final set of hyperparameters used for each of the datasets we implement AGRO on. The exponential weight decay parameter used in greedy G-DRO is set to a default value of 0.5 following prior work and is not tuned.

**Hyperparameter search procedure** : $m$ (the number of groups) is set to 2 times the number of labels in the classification task. This is based on the assumption that one spurious correlation divides training data twice as many groups as the number of labels (groups with and without the feature). A default of 0.2 is used for $\alpha$, following the value that was used for G-DRO on all datasets in Zhou et al. (2021). $T_1$ i.e. the number of epochs used to train the primary model in the first round is set to 3 epochs, following Sanh et al. (2020), where a bias-only model is trained on fewer epochs before using product of experts. This basically amounts to training a weaker ERM model, since we start with a random assignment of groups in the first primary round. The default value for $T_2$ i.e. the number of epochs used to train the adversary, is 1. After one round of training the primary and adversary, we again train the primary model for the same number of epochs that we use to train a strong ERM model baseline that we compare against. This is referred to as N in Table 7. $\theta$ refers to the base pretrained model use for the primary model in each dataset. The hidden-layer size of the two-layer MLP used to instantiate $\phi$, referred to as $\phi_h$ is also tuned. For batch size, we ensure that we use $16 \times m$, to ensure that an almost equal number of examples from each group can potentially get sampled. Note that the minibatch is drawn uniformly from the training data.

**A note on model selection** For the performance gains reported in Tables 2 and 3, model selection is based on the predicted groups, and not known worst group. For model selection, we use the combined performance of $\alpha$-fraction of the worst performing groups discovered by AGRO. When we tune $\alpha$, we are tuning for the same value of $\alpha$ being used in training and validation.

**Statistical significance**    We run AGRO and all other baselines with three different seeds, and report the average worst-group accuracies over these three trials in Table 2 and Table 3.

## A.5    BASELINES

We describe each baseline used in this work in greater detail, including describing choice of base-line specific hyperparameters and implementation details specific to individual tasks, that follows the details in prior work.

**ERM**    The ERM model is trained for $N$ epochs (provided in Table 7). We report on the same batch size as that used in the corresponding AGRO model to draw a fair comparison between them. Learning rate and weight decay are tuned in that order similar to AGRO. Model selection used to report performance in Table 2 and 3 is based on the **entire development set**.

**G-DRO**    This baseline applies for the datasets where spurious correlation based groups are pre-defined (MNLI, Waterbirds, CelebA and CivilComments). We report on a batch size of $6 \times m$, where $m$ is the number of groups and as many epochs as used to train AGRO. We use the value of $\alpha = 0.2$ since it typically corresponds to 1 worst-group for all two class problems and 2 worst-groups for MNLI. We don't use weight decay, and tune learning rate $\eta$ similar to AGRO.

**JTT**    Liu et al. (2021) is a very simple up-sampling strategy where the errors made by ERM are upsampled by training for a second time on those examples. However, hyperparameter tuning is based on access to a known worst group. Instead we do model selection based on the worst-performing group, i.e., the group consisting of all the errors of the ERM model. We tune weight-decay and learning rate $\eta$ similar to AGRO. We start with the same ERM model used for AGRO.

**GEORGE**    Sohoni et al. (2020) generates groups via clustering of ERM model representations and loss-components. However, implementation details for GEORGE showed that only the loss component was used for Waterbirds and CelebA. We follow the same heuristic for MNLI and CivilComments as well. We cluster this loss component via GMM and select number of clusters based on the parameter setting that achieves the highest average per-cluster Silhouette score, following Sohoni et al. (2020). Instead of finding global clusters and overclustering them, we only cluster once (since they didnt find overclustering useful). We use the value of $\alpha = 0.2$ since it typically corresponds to 1 worst-group for all two class problems and 2 worst-groups for MNLI. We tune weight-decay and learning rate $\eta$ similar to AGRO. Model selection is based on the worst-performing group , discovered via the clustering approach. We start with the same ERM model used for AGRO.

**EIIL**    Creager et al. (2021) generates groups via adversarially training a model that learns to maximally violate the Environment Invariance Constraint of Invariant risk minimization (IRM) Arjovsky et al. (2019). However, model selection is based on access to a known worst group. Moreover, implementation details for EIIL showed that a simple heuristic like using labels as groups maximally violates EIC. They find that for large datasets like CivilComments, the convergence time for gradient-based EI increases significantly. Hence they use labels as groups. We also follow this for all datasets except Waterbirds. We use the value of $\alpha = 0.2$ since it typically corresponds to 1 worst-group for all two class problems and 2 worst-groups for MNLI. We tune weight-decay and learning rate $\eta$ similar to AGRO. After group discovery (EI), we use G-DRO for robust optimization. Model selection is based on the worst-performing group , discovered via the grouping strategy. For implementation of EIIL on waterbirds, we follow hyperparamter setting from (Creager et al., 2021).

## A.6    ADDITIONAL EXPERIMENTS

List of additional experiments:

- A.6.1 Experiments on Waterbirds using the ViT(Dosovitskiy et al., 2020) model and on MultiNLI using the Roberta(Liu et al., 2019) pretrained model.
- A.6.2 Stronger ERM model experiments compared to weak models.
- A.6.3 Sensitivity of worst-group performance of AGRO to the hyper-parameters $\alpha$ and $m$.
- A.6.4 Ablation results on waterbirds

| Dataset | | | Known WGs | |
|---|---|---|---|---|
| | | Avg. Acc | WG1 | WG2 |
| MNLI | | | Negation-Contr. | Lexical-Ent. |
| | ERM | 87.00 | 77.34 | 45.33 |
| | G-DRO | 84.48 | 81.65 | 40.12 |
| | **AGRO** | 85.39 | 79.23 | 49.33 |
| Waterbirds | | | WonL | LonW |
| | ERM | 84.23 | 52.63 | 75.10 |
| | GCDRO | 91.40 | 81.03 | 77.51 |
| | **AGRO** | 89.57 | 78.94 | 80.47 |

Table 8: Results on Roberta-base and ViT for MultiNLI and Waterbirds. We report on ERM, G-DRO baselines and AGRO.

| Model | Batch size | Epochs | Avg. Accuracy | WG Accuracy |
|---|---|---|---|---|
| BERT-Base | 32 | 3 | 82.83 | 65.86 |
| Roberta-Base | 32 | 3 | 86.68 | 72.99 |
| | | 10 | 86.16 | 73.38 |
| | | 15 | 86.11 | 73.76 |
| | 128 | 15 | 87.00 | 77.34 |
| Deberta-Base | 128 | 15 | 87.04 | 84.8 |
| Dberta-Large | 128 | 15 | 90.87 | 80.64 |

Table 9: Effect of pretrained model, batch size and training epochs on ERM model's performance on the worst group of MultiNLI

### A.6.1 EXPERIMENTS WITH PRETRAINED MODELS

In Table 8, we report results on using older pretrained models of smaller sizes and less effective pretraining strategies (less data, less effective pre-training tasks). We report on the Roberta-base model (Liu et al., 2019) for MultiNLI and ViT (Dosovitskiy et al., 2020) for Waterbirds.

### A.6.2 EXPERIMENTS WITH ERM MODEL

In Table 9, we report average and worst group accuracies for MultiNLI using different pretrained models and batch sizes. While model size and pretraining initialization improves both average and worst group accuracy, the gap between both does not change.

### A.6.3 EFFECT OF $\alpha$ AND THE NUMBER OF GROUPS

We find that the most important (sensitive) hyperparameters to tune for AGRO are $\alpha$ and number of groups. Figure 3 shows variation of worst-group performance on MNLI as a function of $\alpha$ (left) and $m$ (right). As the proportion of examples in worst groups increases (with $\alpha$), the model converges to the ERM solution. For very small values of alpha, only one group is used for up-weighting, and this group may potentially contain noisy or outlier examples. Consequently, we observe a drop in worst-group performance. A similar pattern also holds with number of groups (keeping $alpha$ fixed to $0.2$). As $m$ increases, groups may contain noisy or outlier examples. If $m$ is very small, groups may not encode any meaningful correlation, thereby reducing to the ERM solution. Note that we use a known worst group on MNLI for model selection for these ablation experiments. Details about other hyperparameters and their values on different datasets and baselines are provided in Appendix A.4.

### A.6.4 FEATURE ABLATIONS ON WATERBIRDS

Results in Table 10.

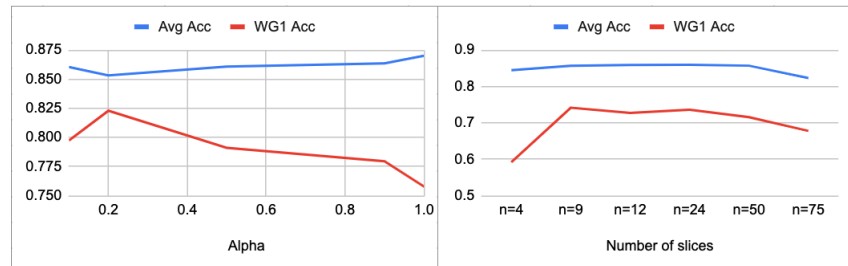

Figure 3: Variance of worst group performance with $\alpha$ and $m$ for MNLI

| Algorithm | AvgAcc | KnownWG |
|---|---|---|
| ERM | 97.66 | 91.73 |
| JTT | 98.33 | 93.98 |
| GEORGE | 98.58 | 95.49 |
| EIIL | 98.25 | 96.99 |
| AGRO | 97.66 | 96.24 |
| G-DRO | 96.25 | 97.74 |

| Features | AvgAcc | KnownWG |
|---|---|---|
| ERM | 97.75 | 90.23 |
| Feature clustering | 94.66 | 63.91 |
| DOMINO clustering | 96.58 | 77.44 |
| AGRO | 98.33 | 94.74 |
| AGRO + pretrained features | 97.66 | 96.24 |

Table 10: Improvement in accuracy on WG1 of MNLI when it is used for model selection (left). Ablation of features in $\mathcal{F}$ and ablation for adversarial learning over clustering (right).

## A.7 HUMAN EVALUATION

We do human evaluation on three datasets (MultiNLI, CelebA and SST2). For each of these dataset, we use AGRO to obtain a large number of groups on the dataset (as opposed to the $m < 25$ groups used for optimization). We do this for qualitative analysis to limit the number of instances in each group to fewer than 100. This allows us to examine the slice in its entirety. We pick the top 5 most underperforming (highest error-rate) groups for all 3 datasets. For task 1 (Coherence estimation), we randomly sample 4 examples from a given group $i$ and 1 additional example from another random group $i'! = i$. We recruit 10 annotators (graduate students familiar working on machine learning and NLP). Each annotator is presented with 3 examples from group $i$, and 2 examples—the 4th example from group $i$ and the randomly chosen example from group $i'$. They are asked to identify which of the two test examples belongs to group $i$. Note that we only show the models' predictions on all examples, since we want to get indirect feedback about model behavior from annotators. We enforce that they provide a definitive choice and explain their reasoning in 1-2 sentences. We solicit 3 different annotations for all 15 slices - a total of 45 different annotations. For the two text classification tasks, we additionally ask annotators to minimally perturb the chosen test example in order to potentially change the model's original prediction (this is not verified during the annotation process). A minimal perturbation includes adding, deleting or substituting a word or 3-4 token phrase in the original instance. Once again, annotators only see model predictions for all examples.

## A.8 QUALITATIVE EXAMPLES OF SLICES

In Figures 4,5 and 7, we present randomly sampled examples from the worst groups discovered by AGRO, which we also used in the human evaluation study

**Premise:** "In the majority of domestic violence cases I see, alcohol, drug and mental health problems are at the heart of the problem.",
**Hypothesis:** "All domestic violence incidents involve alcohol.",
**Predicted Label:** Neutral

**Premise:** "Wayne had the good luck to rule over Westerns, a genre that was set on wide-open land and commemorated the past and dealt in stark moral truths--a genre doomed to grow obsolete, leaving Wayne to dominate the landscape, a proud and lonely warrior.",
**Hypothesis:** "Wayne is one of the very few who dominates the landscape of Westerns. "
**Predicted Label:** Neutral

**Premise:** "but that's my that's probably my main hobby that sewing and reading books that's about it",
**Hypothesis:** "Sewing and reading books are my only hobbies.",
**Predicted Label:** Neutral

**Premise:** Though much of the city has grown up outside the fort modern in time, but not in style 3,000 people live within its walls.
**Hypothesis:** The fort has all of its buildings within the walls.
**Predicted Label:** Neutral

**Qualifiers "only, all, one of the very few" in hypothesis is correlated with the neutral class**

**Premise:** "In Skoptland I came in seventh, but ahead of even Peter King himself.",
**Hypothesis:** "I finished in second place, just behind Peter King."
**Predicted Label:** Contradiction

**Premise:** "If for no other reason, it has provided an excuse to escape for a day the pressure--cooker atmosphere associated with putting the finishing touches on a major rate case decision.",
**Hypothesis:** "It gave them a moment to be relaxed and not to well on the upcoming decisions they needed to make.",
**Predicted Label:** Contradiction

**Premise:** "Use the manual controls.  Hanson waited until he estimated the men who left would be at the controls.",
**Hypothesis:** "Hanson waited for the men to return.
**Predicted Label:** Contradiction

**Premise:** "One exception is the hike north from Olimbos to the shrine to St. John the Baptist at Vrykounda, site of a major island festival on 29 August every year.",
**Hypothesis:** "The shrine to St. John the Baptist at Vrykounda is south of Olimbos."
**Predicted Label:** Contradiction

**Premise and hypothesis contain antonym words, correlated with contradiction**

**Premise:** "and so imagine the sixteen year old yep boy i had a surprise birthday party and a war started in Israel",
**Hypothesis:** "I turned 18 when the war in Israel began.",
**Predicted Label:** Contradiction

**Premise:** "Here you'll find the ruined Abbey of St. Colm, founded in the 12th century and named for St. Columba, who had brought Christianity to western Scotland 600 years earlier.",
**Hypothesis:** "Christianity was brought to Scotland in the 6th century.",
**Predicted Label:** Contradiction

**Premise:** "Helms, who will be 81 when his fifth term ends, is increasingly frail.",
**Hypothesis:** "Helms will turn 91 soon.",
**Predicted Label:** Contradiction

**Premise:** "Nearby is the Monastery of Nea Moni, founded in 1049, and one of the most beautiful Byzantine religious sites in the Aegean.",
**Hypothesis:** "The Monastery of Nea Moni was founded in 1500 and it is very beautiful."
**Predicted Label:** Contradiction

**Premise and hypothesis contain different numerical values or forms, correlated with contradiction**

Figure 4: AGRO slices on MultiNLI. Potential correlation mentioned in purple.

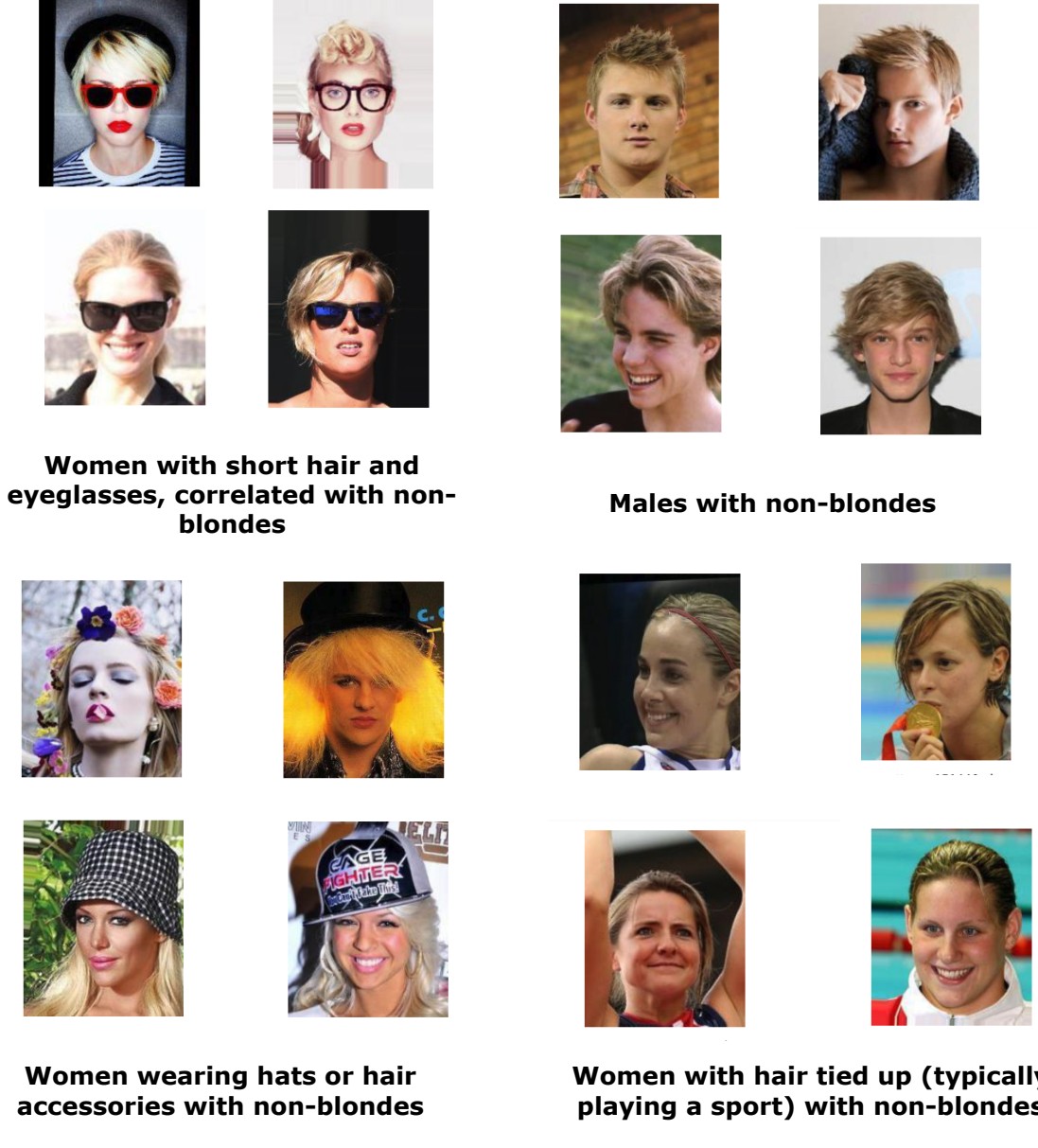

**Women with short hair and eyeglasses, correlated with non-blondes**

**Males with non-blondes**

**Women wearing hats or hair accessories with non-blondes**

**Women with hair tied up (typically playing a sport) with non-blondes**

Figure 5: AGRO slices on CelebA. Potential correlation mentioned mentioned. Model predicts non-blonde on all these examples.

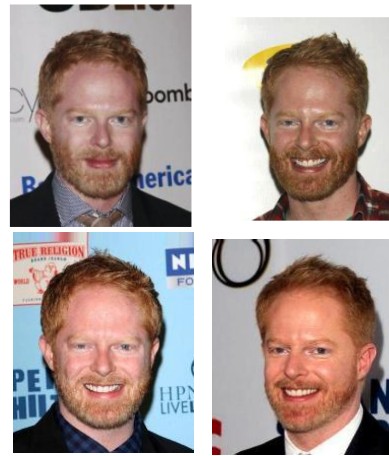

Figure 6: AGRO slices on CelebA with $m = 500$. Large values of $m$ uncover very small groups that can potentially contain noisily labeled examples. For example, this celebrity with ginger hair is labeled as blonde in the dataset, while the model correctly predicts non-blonde and results high error rates for this celebrity.

1) "Rarely has leukemia looked so shimmering and benign ."
Model predicts positive
2) "The experience of going to a film festival is a rewarding one ;
the experiencing of sampling one through this movie is not ."
Model predicts positive
3) "Whereas last year 's exemplary Sexy Beast seemed to revitalize the
British gangster movie , this equally brutal outing merely sustains it ."
Model predicts positive

**The sentence starts off postiive but changes sentiment towards the end.
Positive words associated with positive class**

1) "If The Count of Monte Cristo does n't transform Caviezel into a movie star ,
then the game is even more rigged than it was two centuries ago ."
Model predicts negative
2) "By the time we learn that Andrew 's Turnabout Is Fair Play is every bit as
awful as Borchardt 's Coven , we can enjoy it anyway ."
Model predicts negative
3) "As unseemly as its title suggests .
Model predicts negative

**Negative words - correlated with negative**

1) "This is her Blue Lagoon ."
Model predicts positive
2) "Socrates motions for hemlock ."
Model predicts positive
3) "It uses the pain and violence of war as background material for color .",
Model predicts positive

**Analogies/Comparisons to other entities - correlated with positive**

Figure 7: AGRO slices on SST2. Potential correlation mentioned in purple.

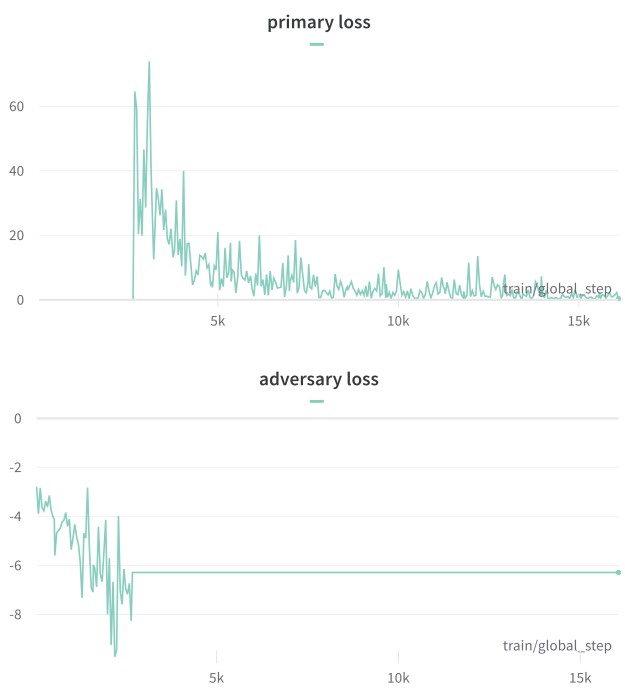

Figure 8: Primary and adversary loss for a round of training AGRO for the MNLI dataset. The primary is trained for T1 iterations and adversary for T2 iterations.

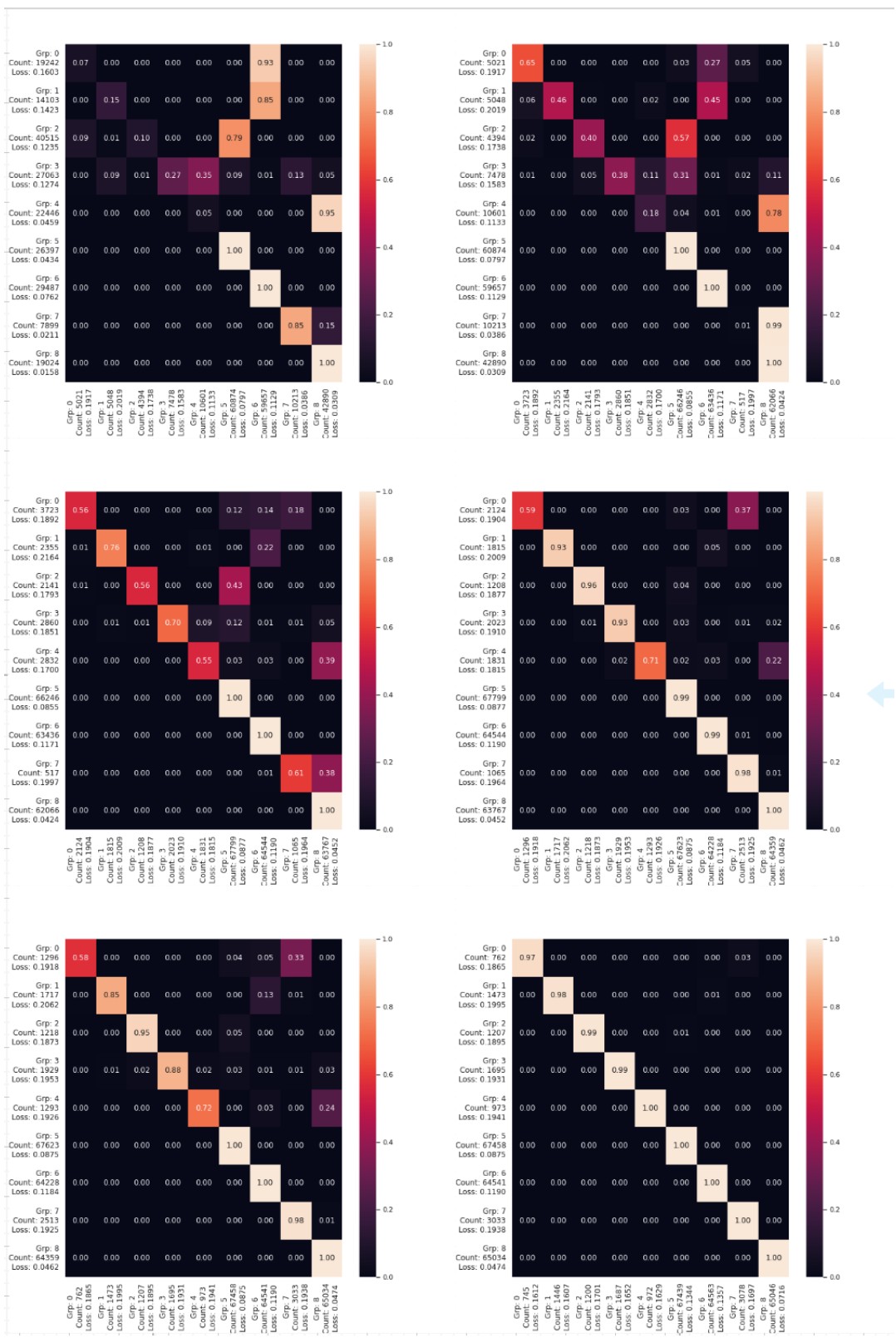

Figure 9: Tracking changes in group identity during the adversary round training. In initial iterations, group indentities change more often, but in later iterations, group identities continue to remain fixed (note concentration on the principal diagonal)

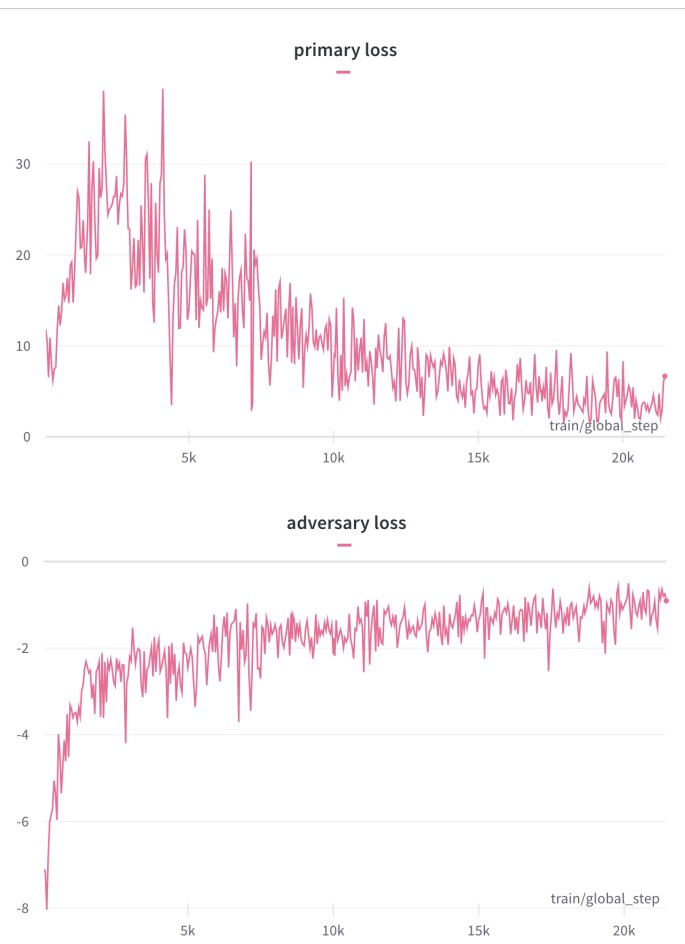

Figure 10: Primary and adversary loss for iterative mix-max optimization i.e. switching between optimizing for $\theta$ and $\phi$ every minibatch. Empirical evaluation results in lower worse-group performance for this optimization vs. switching between adversary and primary after T1 and T2 minibatch iterations.

