# OpenReview forum: "AGRO: Adversarial discovery of error-prone Groups for Robust Optimization"
_ICLR.cc/2023/Conference — ICLR 2023 poster_

### Official Review · Reviewer_eLL5 · 2022-10-19

**Confidence:** 3
**Correctness:** 3
**Technical Novelty And Significance:** 3
**Empirical Novelty And Significance:** 3
**Recommendation:** 6

**Clarity, Quality, Novelty And Reproducibility:**

The paper is clearly written and easy to follow.
I have only a few minor comments.

- In the bottom equation in p.4, $\\hat\\theta_\\mathrm{AGRO}$ should not be an argmax with respect to $\\phi$. The argmax cannot be performed on the argmin (which should be min instead). $\\hat{q}_g$ is not defined.
- In the last part of Section 3.2, the authors explain "we adopt a different initialization for $\\phi$, which is explained in the next subsection", but this should be corrected because it is explained in Section 4.
- In Section 3.3, I did not understand why the authors mention "To estimate model errors on training data, we apply K-fold cross-validation ..." at the first sight because the error estimation does not seem to be used in the proposed method. However, later I figured out that it is indeed used because DOMINO is used as initialization of $\\phi$, which is explained later in Section 4. I guess it's nice to give a pointer in Section 3.3 or aggregate these explanations in a single place to avoid confusion.
- In the caption of Table 2, there should be a space between the first and second sentences.

**Strength And Weaknesses:**

### Strengths

- **Simple method**: The proposed method simply incorporates a new model to predict the underlying group attribute for each data point into the existing worst-case group DRO. This idea itself is simple yet the experimental results demonstrate that this is indeed effective in practice.
- **Thorough experimental evaluation**: The authors provide experiments following several different scenarios such as the supervised (with having access to the group information) and unsupervised (without access to the group information) cases and OOD test. Moreover, they provide a human experiment to see whether the elicited group attributes align with human judgment. Those experiments support the usefulness of the proposed method.

### Weaknesses

- **Rather complicated technical details**: Though the proposed method looks simple, I feel there are several brittle parts in the method. For example, the original formulation to optimize only the worst-group loss is later related to the optimization of the down-weighted loss of the worst $\\alpha$-fraction groups; hence a new hyperparameter $\\alpha$ is introduced. The method seems to heavily depend on the number of groups $m$. The tweaks to the initialization of $\\phi$ (using DOMINO, instead of random initialization) look fragile and require a certain amount of labor. These factors might make the method not easy to handle.

**Summary Of The Paper:**

This paper extends the existing distributionally robust optimization toward the "unsupervised" case, where group information is not given in advance. Following the basic formulation of G-DRO (minimization of the loss function on the worst group), the authors incorporate a grouper model to infer the underlying (unrevealed) group for each instance and perform G-DRO. Despite its simplicity, the proposed method (AGRO) successfully improves the classification performance consistently over ERM, while the existing baselines failed.

**Summary Of The Review:**

The paper proposes an effective solution to the issue that we usually do not have access to the group information in distributionally robust optimization. The simple approach seems pretty effective, hence I feel like accepting this paper.

---

> ### Author Response · Authors · 2022-11-18
> **Response to Reviewer eLL5**
>
> We thank the reviewer for noting the relatively simple yet intuitive idea behind AGRO in predicting worst-case groups in a fully unsupervised setting. As noted by the reviewer, extensive experimental results in unsupervised, semi-supervised, and ood-settings demonstrate that AGRO is indeed effective in practice.
>
> ### **Concerns:**
>
> **Rather complicated technical details:**
> We clarify that the complications noted by the reviewer are actually well substantiated in literature, result in stable learning, and are relatively simple to implement. Despite this complexity, the method can be easily applied to a new task, as we have shown with our OOD experiments. We cite and address concerns raised by reviewer individually.
>
> **“the original formulation to optimize only the worst-group loss is later related to the optimization of the down-weighted loss of the worst alpha-fraction groups;”**
>
> Previous work from Zhou et al. (2021) has shown through extensive experimentation that optimizing a CVaR-style variant of the GDRO objective (down-weighted loss of the worst alpha-fraction groups) performs better than the original formulation. We build off of this improved optimization approach in AGRO. Note that this change does not result in any change in convergence time for GDRO, as noted by Zhou et al. (2021). In the updated version of the paper, we explain this variant in greater detail in Section 3.2 for both, GDRO and AGRO.
>
> **“hence a new hyperparameter $\alpha$ is introduced.”, “The method seems to heavily depend on the number of groups”**
>
> AGRO requires tuning the hyperparameter $\alpha$ (fraction of groups that are up- or down-weighted in optimization) and the number of groups ($m$). We contend that the number of groups hyperparameter is integral to the general problem of identifying worst-groups and spurious features in a *fully unsupervised* approach, since a fundamental challenge in this problem is that the number of worst groups is not known apriori. Also, the hyperparameter  $\alpha$ is already employed in the original GDRO optimization (see Oren et. al, Zhou et. al.) and not introduced in AGRO. In accordance with this, AGRO is therefore sensitive to the choice of $m$ and $\alpha$ but fairly robust to choices of other hyperparameters.
>
> **“The tweaks to the initialization of $\phi$ (using DOMINO, instead of random initialization) look fragile and require a certain amount of labor.”**
>
> Training the grouper (adversary) model $\phi$ to be initialized to DOMINO predictions is relatively low latency. DOMINO code (https://github.com/HazyResearch/domino) to get groups is used directly for our application and takes < 10 minutes to run on any of our evaluation datasets on a 16-core CPU. Furthermore, training a 2-layer MLP to predict DOMINO groups also takes < 10 minutes on a Tesla K80.
>
> ### **Suggested Edits:**
> Thank you for your suggestions about the writing. We have incorporated your suggestions in the draft. A summary of relevant edits is given below:
>
> **“In the bottom equation in p.4,  $\hat{\theta}_{AGRO}$ should not be an argmax with respect to  $\phi$. The argmax cannot be performed on the argmin (which should be min instead). $\hat{q}_g$ is not defined.”**
>
> We have updated the notation in addressing concerns from Reviewer YtmE, removing $\hat{q}_g$  and changing the order of argmin and argmax. Note that equation (1) represents the joint optimization to find *parameters* $\phi$ and $\theta$ in independent rounds (see Algorithm 1). Hence we use argmin instead of min. The argmax is with respect to the adversary parameters  $\phi \in \Phi$ and the inner argmin is with respect to the task parameters $\theta \in \Theta$.
>
> **“In Section 3.3, I did not understand why the authors mention "To estimate model errors on training data, we apply K-fold cross-validation ..." at the first sight because the error estimation does not seem to be used in the proposed method. However, later I figured out that it is indeed used because DOMINO is used as initialization of $\phi$, which is explained later in Section 4. I guess it's nice to give a pointer in Section 3.3 or aggregate these explanations in a single place to avoid confusion.”**
>
> Indeed, DOMINO is used to initialize $\phi$ as described in Section 4 and we have added a backpointer to the initial description of DOMINO in Section 3.1 to address this.
> However,  the K-fold cross-validation trick described in Section 3.3 is used to estimate prediction probabilities on training data $p(y_i|x_i;\theta_k)$, which are part of the feature vector $f_i \in \mathcal{F}$. This feature vector is used as input to the adversary model $\phi$ when training AGRO. Since this is the primary motivation for using K-fold cross-validation trick to generate input features for $\phi$, we have included its description in Section 3.3, where we detail how $\mathcal{F}$ is defined in AGRO.
>
> **Typos**
> We address all errors pointed out by the reviewer in the updated version of the paper.

---

> > ### Comment · Reviewer_eLL5 · 2022-11-25
> > **Reply**
> >
> > I appreciate the authors for addressing the comments dedicatedly. Although I still have a concern about the hyperparameter sensitivity, I understand $m$ is almost unavoidable in this problem setting. I found Figure 3 in the appendix shows robustness against the hyperparameter choices to some extent, which supports the claim fairly. Generally speaking, I like the overall simple idea.

---

### Official Review · Reviewer_YtmE · 2022-10-21

**Confidence:** 3
**Correctness:** 3
**Technical Novelty And Significance:** 2
**Empirical Novelty And Significance:** 3
**Recommendation:** 5

**Clarity, Quality, Novelty And Reproducibility:**

Novelty: The use of a parametrized grouper model, although not entirely new (Lahoti et al. already do something similar in the context of example re-weighting) is a nice addition to the group DRO literature, and the experimental evaluations are elaborate.

Quality: My main concern is about the technical correctness of the algorithm use for the specific formulation the authors care about.

Clarity: The notations need to be made more precise. The algorithmic detail need to be explained in a more self-contained manner.

**Strength And Weaknesses:**

**Pros:**
- The problem of worst-group optimization under unknown groups is important, and tackling this problem with no prior knowledge of the group identities makes the proposed approach practically relevant
- Setting up an adversarial game between a parametrized grouper model and  the DRO subroutine is a natural idea to try
- The authors carry out an extensive experimental comparison with many recent baselines and multiple benchmarks and human eval tasks.

**Cons:**

My main concern is the lack of sufficient explanation or justification for why the algorithm proposed does indeed optimize the objective that the authors seek to optimize. It appears the authors directly borrow an algorithm from Oren et al. (2019)  for their inner G-DRO solver, but its unclear if the DRO objective in Oren et al. is the same as the one in the present paper. Moreover, the details of the algorithm (both in the main text and appendix) are left vague and one needs to read up prior papers to understand the workings of the algorithm.

**Mismatch in objective between algorithm and formulation:**

My understanding is that the inner group DRO in the formulation on page 4 seeks to minimize over $\theta$, the following worst-group objective:

$$
\max_{g \in \mathcal{G}} \mathbb{E}_{(x, y) \sim q_g}\left[ \ell(x, y; \theta) \right],
$$

where if I understand correctly (although the notation is a bit imprecise), $q_g$ is a distribution over instances belonging to group $g$ and is the output of a grouper neural network.

To solve this inner problem, the authors employ an algorithm from Oren et al., which however seeks to optimize a slightly different CVaR-style objective (as elaborated in Zhou et al. (2021)):

$$
\max_{q \in \mathcal{Q}} \mathbb{E}_{g \sim q,~ (x, y) \sim p(x, y|g)}[ \ell(x, y; \theta) ],
$$

where $\mathcal{Q}$ =  { $q: q(g)  \leq p_{train}(g) / \alpha$ } is a specific uncertainty set over groups defined for a particular fraction $\alpha$.

In the present paper, the formulation presented does not construct this uncertainty set, and seeks to directly optimize over all groups in $G$. The parameter $\alpha$ is however present in algorithm and treated as a hyper-parameter in their experiments.

I think it is important that the authors clarify this mismatch between the formulation they present and eventual algorithm they use.

It would also be helpful if the authors can be *more precise with their notation*, whether it is in formulating the distribution $q_g$ (was not initially clear if this was over examples or groups), or in the description of the individual steps in the algorithm, such as the use of an exponentiated moving average EMA (not entirely clear what "samples of each group in B" means in Alg 1).

**Convergence of Algorithm 1**:

Another smaller concern relates to the convergence of the alternating optimization between the adversary and the G-DRO. Solving min-max problems can be tricky and may lead to oscillations if the updates are not carefully chosen. For example, merely having the max-player perform a full maximization over its parameters and the min-player perform a full-minimization at each round generally is not known to have good convergence behavior. Limiting each player to few minibatch gradient updates, as done in this paper, may be a useful practical trick, but I think it is also important that authors at least provide some citations for why the approach taken would lead to convergence.

Minor question:
In your implementation of G-DRO from Sagawa et al., did you implement the per-group regularized updates in eq (5) (which I think is what they use in their experiments), or the online version in Algorithm 1 in their paper?

Additional related literature:
Kirichenko et al. Last Layer Re-Training is Sufficient for Robustness to Spurious Correlations. ArXiv:2204.02937.



**Summary Of The Paper:**

The presence of spurious correlations between the labels and features can be an impediment to out-of-sample generalization. A common solution in the literature for this problem is apply a group distributionally robust optimization (G-DRO) to minimize the worst-loss over subgroups in the data. However, prior approaches for group DRO either require the group identities to be known in advance either in the training or validation set, or require some task-specific knowledge.

This paper presents combines the discovery of error-prone groups with the G-DRO procedure into a single end-to-end training procedure by setting up an adversarial game between a G-DRO solver and an adversary who seeks to identify soft group assignments to maximize worst-group error. The adversary is a parameterized neural network that outputs a distribution over training examples for each group.



**Summary Of The Review:**

Given the mismatch between the algorithm and the formulation, the paper falls short of the acceptance threshold. I am happy to revisit my score after hearing back from the authors.

---

> ### Author Response · Authors · 2022-11-18
> **Response to reviewer YtmE**
>
> We thank the reviewer for noting the practical relevance of the problem of worst-group optimization with no prior knowledge of the group identities. We also thank the reviewer for noting that setting up an adversarial game between a parametrized grouper model and the DRO subroutine is a natural and intuitive idea.
>
> ### **Concerns**
>
> **Mismatch in objective between algorithm and formulation:**
>
> We clarify the mismatch between Algorithm 1 and the formulation presented in Equation (1) in 3.2.
>
> Primary Round (inner GDRO optimization): We do implement the CVaR-style objective (as elaborated in Zhou et al. (2021)) and have updated the paper accordingly. The uncertainty set, as noted by reviewer, is
> $Q$ = { $q: q(g)  \leq p_{train}(g)/\alpha$} , where  $p_{train}$ is the empirical distribution of groups and $p_{train}(g)$  is the empirical probability of group g in the training data.
>
> Zhou et al. (2021) use  $p_{train}(g) = \frac{n_g}{|D|}$ where $n_g$ is the number of examples of group $g$ in training data D. i.e. $p_{train}(g)$ is simply the proportion of examples belonging to group $g$. In implementing minibatch training, $n_g$ is estimated based on the number of examples of group $g$ in minibatch (of size B). In order to have an unbiased estimate for  $p_{train}(g)$, they maintain an expected moving average $p^*_{train}(g)$. Thus,
> $$
> p^*_{train}(g) \leftarrow \text{EMA}(\text{samples of group g in B}, p^*_{train}(g))
> $$
>
> In our setting (AGRO), $p_{train}(g)$ is computed based on a group assignments probabilities output by the adversary model ($\phi$) trained in the previous round. Specifically,
> $$
> p^*_{train}(g) \leftarrow \text{EMA}( \sum_{j=1}^{|B|} \frac{p(g|f_j, \phi^{(t-1)})}{|B|}, p^*_{train}(g))
> $$
>
> where $p(g|f_j, \phi^{(t-1)})$ is the probability of example $j$ belonging to group $g$ in the minibatch, which is estimated by the adversary model ($\phi^{(t-1)}$).
>
> We change equation $\theta_{AGRO}$ so that it reflects more accurately what is detailed in Algorithm 1:
>
> $\phi^*, \theta^*_{AGRO} = arg\min_{\theta \in \Theta}\( arg\max_{\phi \in \Phi}\( \hat{R}(\theta) := \max_{q \in Q(\phi)} \mathbb{E}_{g \sim q, x,y \sim p(x,y|g)} [l(x, y; θ)] \)\)$
>
> The uncertainty set Q is defined as in Zhou et. al., with one main difference:  $p_{train}(g)$ is not fixed, but estimated by the model $\phi$. Specifically, $p_{train}(g)$  is computed based on $p(g|f_j, \phi^{(t-1)})$.
>
> *Regarding clarity and self-contained explanation of the algorithmic details:*
> We acknowledge that due to a lack of space, specific details and changes made to the CVaR-style optimization were not discussed in detail in the main body of the paper (with details moved to Appendix A.2). More details about the exact optimization procedure used are in that section (which has also been edited to be consistent with changes in the main body). Upon reviewer's suggestion, we can move certain parts back to the main body.
>
> **Stability of training**
>
> In the primary round where GDRO optimization occurs, we find that the model converges when all groups have similar losses. In the adversary round, since we fix the task model parameters $\theta$, we find that the group memberships change frequently initially, but after a few iterations (<T2) the memberships do not change. To show this, we have added training loss plots (Appendix, Figure 8) for primary and adversary optimization rounds. We have also visualized the change in group memberships across several iterations of the adversary training round (Appendix, Figure 9). These analyses are done for MNLI and authors note that the same patterns persisted on other tasks as well.
>
> In A.2, we also explain in detail how we achieve stable training for AGRO: In particular, the sections titled *Implementation details for AGRO* and *Initializing $\phi$* should be noted. Briefly,
>
> - We initialize the grouper model $\phi$ to predict DOMINO error-aware groups in order to ensure that the optimization does not lead to trivial degenerate solutions i.e. all examples being assigned to 1 group.
> - Empirically, we found that training the primary and adversary independently for a few minibatches (T1, T2 respectively) is more stable and converged faster than iterating between training the adversary and primary every minibatch.
> - We maintain EMA of group losses which are then used to make updates to the model. This was helpful in stabilizing the training procedure and has been adopted in prior work (Zhou et. al.).
>
> **Answer to clarification question**
> In implementing G-DRO from Sagawa et al., we use the CVaR-style objective (as elaborated in Zhou et al. (2021).
>
> **Citation Edits**
> We have added the suggested citation.

---

### Official Review · Reviewer_UiKr · 2022-10-22

**Confidence:** 3
**Correctness:** 3
**Technical Novelty And Significance:** 2
**Empirical Novelty And Significance:** 3
**Recommendation:** 5

**Clarity, Quality, Novelty And Reproducibility:**

This paper is good in clarity and quality. The proposed method is somewhat novel. Experimental details are provided for reproduction.

**Details Of Ethics Concerns:**

No ethics concerns appear.

**Strength And Weaknesses:**

Strength:
- This paper is well-organized and easy to follow. The general structure is quite clear.
- The proposed method does not need the group annotation and can be trained in an end-to-end manner, which has great applicability.
- The adversarial discovery for groups is a novel concept in DRO. Intuitively, the selected worst group can be highly informative for the current loss landscape, thus being helpful for learning a robust model.

Weakness:
- As is well-known that adversarial process is highly unstable and hard to train. This paper incorporative another adversarial round into DRO which already contains an adversarial component. So, I am doubtful for the training stability and its convergence.
- The main contribution of this paper is the adversarial discovery-based grouper. However, there is no effectiveness analysis on the grouper performance. Since the group distribution is modeled by a continuous probability, I am curious about the softmax accuracy of the grouper on clustering the groups. How does the clustering accuracy change along the training process?
- The proposed method employs the error slice discovery technique into the DRO setting. However, the concept is slice between group is not clearly described. From my understanding, each group could contain different slices. For example, blond woman is a group, in which there exists two slices: blond woman with hat and blond woman without hat. So, it is feasible to directly employ the DOMINO method to conduct grouping? It is possible that the group only helps the performance of worst-slice instead of worst-group?

**Summary Of The Paper:**

This paper studies the distributionally robust optimization problem. Due to existing methods depending on expensive group annotations, the proposed AGRO method proposes to discover different groups via an adversarial opponent. Specifically, there are two models are designed in the AGRO, namely the task model and the grouper model. The task model minimizes the worst-case risk, while the grouper model maximizes the same risk. As a result, the grouper model is shown to be effective in distinguishing different groups without the group prior. By conducting extensive experiments on image and language datasets, the AGRO is shown to be superior to compared baseline methods.

**Summary Of The Review:**

I have carefully read the whole paper. This paper makes some contribution to DRO by introducing adversarial discovery, however, there are still some concerns (see weaknesses). If the authors can address my concerns, I will consider raising my score.

---

> ### Author Response · Authors · 2022-11-18
> **Response to Reviewer UiKr**
>
> We thank the reviewer for highlighting the broader applicability of AGRO in scenarios where expensive group annotations are not available and its end-to-end optimization procedure. We also thank the reviewer for noting that adversarially finding a group assignment that can be highly informative for the current loss landscape is intuitive.
>
> ### **Concerns:**
>
> **1. Training stability and convergence of AGRO**
>
> A note on the innermost max(finding the worst-case loss): As noted by the reviewer, the adversarial component of DRO involves tractably computing worst-case loss over an uncertainty set of distributions over training data $\mathcal{Q}$ (see Duchi et al., 2019 and updated Section 3.1/3.2).
> G-DRO simplifies this by modeling the uncertainty set as discrete groups, so the worst-case expected loss is simply the largest group-loss value (as explained in detail in Sagawa et. al., 2019).
>
> Stability of min-max optimization of the worst-case loss (AGRO): In the primary round, where worst-case loss is minimized, we find that the model converges when all groups have similar losses. In the adversary round where worst-case loss is maximized, we find that the group memberships change initially but after a few iterations (<T2), the memberships do not change and loss converges to a large value (note that we fix the task model parameters $\theta$ in this round). To demonstrate this, we have added training loss plots for the primary and adversary optimization rounds  (Appendix, Figure 8). We have also added a visualization of the change in group memberships across epochs for several iterations of the adversary training (Appendix, Figure 9). These analyses are done for MNLI and authors note that the same patterns persisted on other tasks as well.
>
> In A.2, we also explain in detail how we achieve stable training for AGRO: In particular, the sections titled (a) *Implementation details for AGRO* and (b) *Initializing $\phi$* should be noted. We can move this to the main body of the paper if the reviewers suggest so. Briefly,
> - We initialize the grouper model $\phi$ to predict DOMINO error-aware groups in order to ensure that the optimization does not lead to trivial degenerate solutions i.e. all examples being assigned to 1 group.
> - Empirically, we found that training the primary and adversary independently for several minibatches (T1, T2 respectively) is more stable and converged faster than iterating between training the adversary and primary every minibatch.
> - We maintain exponential moving average (EMA) of group losses that are used to sort groups and then make updates to parameters as shown in Algorithm 1.  This was helpful in stabilizing the training procedure and has been adopted in prior work (Zhou et. al.).
>
> **2. Analysis of the grouper performance**
>
> It is not feasible to measure clustering accuracy because the grouping corresponds to potentially unknown spurious correlations and hence we do not have gold clusters. In Figure 4 and Figure 5 in the Appendix, we qualitatively show that AGRO re-discovers known spurious correlations in its predicted groups along with previously uncharacterized correlations. We measured grouper accuracy on known correlations in order to ascertain how accurate this re-discovery is. We find that for MNLI, we find a group that has 63% examples of negation with entailment and neutral labels. For CelebA, we find a group that has 72% examples of males with blonde hair which were mislabeled as non-blondes. Furthermore, human analyses on discovered groups in Section 6 and Table 7 highlight that humans find discovered groups to be coherent with respect to a human-understandable concept or pattern.
>
> **3. Misunderstanding about Slice vs Group terminology and contributions over DOMINO**
>
> We want to clarify that Slice and Group are the same concept in our work, and apologize if these terms were used interchangeably in the writing. To clarify, blonde women with hats is a group (or slice) discovered by AGRO in Figure 1. This group is a worst-group since the ERM model is predicting non-blonde while the ground truth label for the images is blonde.
>
> Furthermore, we note that the novelty of AGRO is substantially greater than just employing the error slice discovery technique in the GDRO setting. While error slice discovery techniques like DOMINO also potentially identify spurious features in the groups they discover, they have not as yet been employed to actually improve model robustness.
> In fact, DOMINO groups when used for GDRO directly perform worse compared to AGRO-groups (see ablations in Table 4 (right). We hypothesize that AGRO finds groups that are most informative for GDRO, unlike techniques like DOMINO.

---

### Official Review · Reviewer_uGUw · 2022-10-29

**Confidence:** 4
**Correctness:** 4
**Technical Novelty And Significance:** 3
**Empirical Novelty And Significance:** 4
**Recommendation:** 8

**Clarity, Quality, Novelty And Reproducibility:**

I have no qualms about clarity and quality. The approach is clearly novel, and a step in the right direction.

There are several knobs at play here, and as long as the authors commit to sharing their code online, I have no concerns about reproducibility.

**Strength And Weaknesses:**

Strengths:

--- As far as I know, this is the first method that discusses automatic groupings as part of the optimizing process, as opposed to assuming known groupings like in some previous works. This is novel and a significant step in the right direction imo.
--- The paper is very-well written, and well-placed in the known literature. The authors discuss the motivation for their method and illustrate the novelty and the impact well. The authors mention that their work is motivated as a combination of two previous known works, and while this is true I think the overall novelty extends beyond that.
--- The empirical evaluations are adequate. The authors have shown competitive performance across several datasets, and have done useful ablation and human subject evaluations to illustrate the effectiveness of the method.

Weaknesses:
---  While this is also done in previous works, I am somewhat confused by the goal of optimizing for the "worst-case" loss. If there are outliers, would such an approach not exacerbate the problem?



**Summary Of The Paper:**

This paper proposes a method to simultaneously infer soft groupings as well as optimizing for worst performance within the group in an iterative minimizing-maximizing optimization routine. The authors show performance gains on datasets with known and unknown spurious correlations as well as do ablation analysis and human-subject mechanical turk like experiments to show that the method performs competitively.

**Summary Of The Review:**

I think the paper is strong, it makes useful contributions in terms of automatic soft groupings and the corresponding empirical evaluations that ratify the utility of the proposed method. I would recommend accepting the paper.

---

> ### Author Response · Authors · 2022-11-18
> **Response to Reviewer uGUw**
>
> We thank the reviewer for recognizing the broader novelty of our work that extends beyond the combination of GDRO and slice discovery methods - specifically, an iterative minimizing-maximizing optimization routine that simultaneously infers soft groupings and improves worst group performance. We thank the reviewer for noting that our work on robust optimization with unknown spurious correlations is a significant step in this direction.
>
> **Question on whether optimizing worst-case loss may cause outliers to be detected:**
>
> This is possible, especially for large values of m (the number of groups). For example, if m is very large, a worst-group discovered by AGRO may have very few examples that often correspond to label noise. We mitigate this by explicitly tuning m. For example, setting m to a high value like 500 in CelebA gives us a group that corresponds to annotation noise (see the updated Figure 6 in Appendix). Notably, we tune m between small values of 4-50. For a particular dataset, we only sweep over values of m such that there are $>5\%$ training examples on average in a group. The values that are finally used to report results for AGRO (Table 7) range from 6-24. Hyperparameter tuning details for m are described in greater detail in A.4 and Table 6 in Appendix. Figure 3 in Appendix also shows how known worst group accuracy actually peaks at m=9 for MNLI and then decreases, underscoring that our hyperparameter search strategy is effective at avoiding the detection of outliers.

---

### Author Response · Authors · 2022-11-18
**Author Response Summary**

We would like to thank all reviewers for their thorough and constructive comments. All reviewers highlight the importance of tackling the problem of robustness to unknown spurious correlations, the general applicability and novelty of our approach, and our extensive empirical validation of the approach.
Reviewers find the adversarial minimization-maximization optimization routine for worst-group discovery and worst-group robustness intuitive. Reviewers also noted the consistent performance gains of AGRO on known and unknown spurious correlations, useful ablations, and informative human subject evaluations.

We address the following common concerns raised by reviewers, with more details in individual responses. We briefly mention updates made in the revision in each case:

**Mismatch of Algorithm 1 and Equation 1 (Reviewer YtmE ‘s concerns)**

We clarify the misalignment between Algorithm 1 and the objective in Equation 1 in greater detail in the individual response to Reviewer YtmE .
Briefly, we edit Equation 1 so that it reflects more accurately what is detailed in Algorithm 1. Equation 1 still presents the same min-max optimization, but now best represents the CVaR-like objective that was implemented.  Reviewer YtmE also had questions about changes made to CVaR-style optimization for AGRO in Algorithm 1, which are clarified and reflected in the update to Algorithm 1. A discussion about Equation 1 and additional implementation details in Algorithm 1 have also been added to Appendix A.2.

**Stability of training AGRO:**

Reviewers UiKr, YtmE raise concerns about training stability of the min-max optimization procedure of the adversary and G-DRO models.  We have added training loss plots (Appendix, Figure 8) for the adversary and primary optimization rounds to visualize training stability.

Convergence: In the primary round, where worst-case loss is minimized, the model converges when all groups have similar losses. In the adversary round where worst-case loss is maximized, we find that the group memberships change initially but after a few iterations (<T2) the memberships do not change and loss converges (note that we fix the task model parameters $\theta$ in this round).  We have also added a visualization of the change in group memberships across several iterations of the adversary training round (Appendix, Figure 9). These analyses are done for MNLI and authors note that the same patterns persisted on other tasks as well. In A.2, we also explain in detail how we achieve stable training for AGRO, which is clarified further in individual responses to reviewers.

**Hyperparameter search and complexity:**

Reviewer eLL5 has questions about sensitivity of AGRO to the hyperparameters $\alpha$ (fraction of groups that are up- or down-weighted in optimization) and number of groups ($m$). We contend that the number of groups hyperparameter (m) is integral to the general problem of identifying worst-groups and spurious features in a fully unsupervised approach since a fundamental challenge in this problem is that the number of worst groups is not known apriori. We add Figure 6 to Appendix to demonstrate the effect of having a very large value of m. We note that hyperparameter $\alpha$ is already employed in the original GDRO optimization and not introduced in AGRO.

---

### Decision · Program_Chairs · 2023-01-20

**Decision:**

Accept: poster

**Justification For Why Not Higher Score:**

Although the algorithm empirically is effective, it has multiple knobs and no performance guarantees.

**Justification For Why Not Lower Score:**

Reviewers agree the method is a nice addition to the GRO literature.

**Metareview: Summary, Strengths And Weaknesses:**

Group distributionally robust optimization (GDRO) mitigates degradations in accuracy due to distributional shifts caused by the presence of spurious correlations in the training data by minimizing the worst expected loss over pre-identified groups in the training data. In practice such groups as seldom known a priori; this paper tackles the problem of simultaneously finding these groups and minimizing the loss over them to achieve GDRO by posing the problem as a min-max optimization problem. The reviewers expressed some concern over the lack of discussion and guarantees of convergence of the algorithm given for solving the min-max problem, but overall find the use of a parameterized grouper to be a nice addition to the GDRO literature, and the method to be novel, simple, and empirically effective.

**Note From Pc:**

if the above contains the word "oral" or "spotlight" please see: "oral" presentation means -> notable-top-5% and "spotlight" means -> notable-top-25%. As stated in our emails, we are disassociating presentation type from AC recommendations